# Genetic diversity of *Salmonella enterica* isolated over 13 years from raw California almonds and from an almond orchard

**Anne-laure Moyne**[1,2], **Opeyemi U. Lawal**[3], **Jeff Gauthier**[4], **Irena Kukavica-Ibrulj**[4], **Marianne Potvin**[4], **Lawrence Goodridge**[3,5], **Roger C. Levesque**[4], **Linda J. Harris**[1,2]*

**1** Department of Food Science and Technology, University of California, Davis, California, United States of America, **2** Western Center for Food Safety, University of California, Davis, California, United States of America, **3** Canadian Research Institute for Food Safety, Department of Food Science, University of Guelph, Guelph, Ontario, Canada, **4** Institut de biologie intégrative et des systèmes (IBIS), Faculté de médecine, Université Laval, Québec, Québec, Canada, **5** Food Safety and Quality Program, Department of Food Science and Agricultural Chemistry, McGill University, Sainte Anne de Bellevue, Quebec, Canada

* ljharris@ucdavis.edu

**Data Availability Statement:** All the data are in the NCBI database with the following links: https://www.ncbi.nlm.nih.gov/bioproject/PRJNA941918

## Abstract

A comparative genomic analysis was conducted for 171 *Salmonella* isolates recovered from raw inshell almonds and raw almond kernels between 2001 and 2013 and for 30 *Salmonella* Enteritidis phage type (PT) 30 isolates recovered between 2001 and 2006 from a 2001 salmonellosis outbreak-associated almond orchard. Whole genome sequencing was used to measure the genetic distance among isolates by single nucleotide polymorphism (SNP) analyses and to predict the presence of plasmid DNA and of antimicrobial resistance (AMR) and virulence genes. Isolates were classified by serovars with Parsnp, a fast core-genome multi aligner, before being analyzed with the CFSAN SNP Pipeline (U.S. Food and Drug Administration Center for Food Safety and Applied Nutrition). Genetically similar (≤18 SNPs) *Salmonella* isolates were identified among several serovars isolated years apart. Almond isolates of *Salmonella* Montevideo (2001 to 2013) and *Salmonella* Newport (2003 to 2010) differed by ≤9 SNPs. *Salmonella* Enteritidis PT 30 isolated between 2001 and 2013 from survey, orchard, outbreak, and clinical samples differed by ≤18 SNPs. One to seven plasmids were found in 106 (62%) of the *Salmonella* isolates. Of the 27 plasmid families that were identified, IncFII and IncFIB plasmids were the most predominant. AMR genes were identified in 16 (9%) of the survey isolates and were plasmid encoded in 11 of 16 cases; 12 isolates (7%) had putative resistance to at least one antibiotic in three or more drug classes. A total of 303 virulence genes were detected among the assembled genomes; a plasmid that harbored a combination of *pef*, *rck*, and *spv* virulence genes was identified in 23% of the isolates. These data provide evidence of long-term survival (years) of *Salmonella* in agricultural environments.

https://www.ncbi.nlm.nih.gov/bioproject/PRJNA951760

**Funding:** LJH 18-HarrisL-AQFSS-01 Almond Board of California (https://www.almonds.com) LG/RCL Genome Canada (https://genomecanada.ca) and Genome Quebec (https://www.genomequebec.com/en/home/) The funders had no role in study design, data collection and analysis, decision to publish, or preparation of the manuscript.

**Competing interests:** The authors have declared that no competing interests exist.

## Introduction

From 2001 to 2006, *Salmonella enterica* serovar Enteritidis was implicated in three outbreaks linked to raw almond consumption. Epidemiologic and traceback investigations of a 2000 to 2001 (denoted 2001) salmonellosis outbreak in Canada and in the United States identified a rare phage type (PT) of *Salmonella* Enteritidis, PT 30, from clinical samples, raw almonds sampled at retail, environmental (swab) samples at an almond processor and a huller-sheller facility, and environmental drag swabs obtained from multiple orchards in California [1]. *Salmonella* Enteritidis PT 30 was recovered in 2001 (three of six samples collected) [1]) and, in a subsequent study, at least two times in each year between 2002 and 2006 from environmental drag swabs collected in one of these orchards (eight to 96 samples collected each year) [2]. Raw almonds were epidemiologically linked to clinical cases of *Salmonella* Enteritidis PT 30 reported in Sweden in 2005 to 2006 (denoted 2006) [3] but *Salmonella* was not isolated from the implicated California almonds. Between 2003 and 2004 (denoted 2004), another rare phage type, *Salmonella* Enteritidis PT 9c, was linked to consumption of raw California almonds in the United States [4].

Excluding the farms linked to the 2001 outbreak, the prevalence and levels of *Salmonella* were determined in a multi-year survey of raw almond kernels. Samples of individual lots of raw almond kernels were collected as they were received by handlers (processors) located throughout the almond-growing regions of California from 2001 to 2007, and in 2010 and 2013 [5–7]. Inshell almonds were included in the survey in 2006 and 2007 [6]. The prevalence of *Salmonella* in 14,949 lots of raw almond kernels was 0.98% ± 0.29% over the 9 years (146 positive 100-g samples) [7]. Levels of *Salmonella* were estimated for 118 of the positive lots; mean and median levels of *Salmonella* were 1.14 ± 1.69 and 0.79 most probable number (MPN)/100 g, respectively, with single lots at 9.25 and 15.4 MPN/100 g [7]. Of the small number of raw inshell almond lots (455) evaluated in 2006 and 2007, 1.5% (seven 100-g subsamples) were positive [6]. Using classical serological nomenclature, the *Salmonella* isolates retrieved from these surveys were serotyped into 45 different serovars, including *Salmonella* Enteritidis PT 30 and PT 9c [5, 6].

Pulsed-field gel electrophoresis (PFGE), multilocus variable-number tandem repeat analysis (MLVA), and comparative genomic indexing (CGI) were applied to *Salmonella* Enteritidis strains associated with the 2001, 2004, and 2006 almond outbreaks, including clinical and environmental isolates [8]. The *Salmonella* Enteritidis PT 30 and PT 9c strains could be separated from each other and from other *Salmonella* Enteritidis phage types based on DNA enzyme restriction profiles, MLVA types, and genes identified by CGI. However, neither PFGE nor MVLA could discriminate among the *Salmonella* Enteritidis PT 30 isolates associated with the 2001 and 2006 almond-associated outbreaks. *Salmonella* Enteritidis PT 30 almond-associated outbreak strains could not be distinguished from epidemiologically unrelated *Salmonella* Enteritidis PT 30 clinical strains included in the study [8]. Among *Salmonella* Enteritidis PT 30 strains with identical genotypes, metabolic analyses with Biolog revealed differences between clinical and environmental isolates [8], indicating the discriminatory limit of the then current genotyping methods. Environmental *Salmonella* Enteritidis PT 30 isolates, collected between 2001 and 2006 from one of the 2001 outbreak-associated orchards, clustered in two groups based on the separation by PFGE of their XbaI-digested DNA [2].

Whole genome sequencing (WGS) has replaced PFGE for investigating foodborne outbreaks because of its higher resolution, and this methodology has been incorporated into routine public health surveillance since 2014 [9, 10]. Clonality of pathogen strains determined by WGS analyses can provide information about contamination during food production and distribution [11]. Different workflows have been developed to assess levels of genetic relatedness

of foodborne pathogens by the National Center for Biotechnology Information (NCBI) with the Pathogen Detection Pipeline (https://www.ncbi.nlm.nih.gov/pathogens/), the U.S. Food and Drug Administration with the Center for Food Safety and Applied Nutrition CFSAN SNP Pipeline [12], and the U.S. Centers for Disease Control and Prevention with Lyve-SET [13], based on the specific needs of the respective agencies. The CFSAN SNP Pipeline creates high quality SNP matrices from WGS data that allow connection between clinical isolates to food or environmental isolates based on their evolutionary relationship [12]. Resident pathogen strains in facilities will have closely related WGS profiles, whereas transient pathogen strains will have unique or unrelated WGS profiles [11, 14].

The objective of the present study was a comparative genomic analysis of 171 *Salmonella* (45 serovars) isolated from raw inshell almonds and almond kernels in 9 years of surveys conducted between 2001 and 2013; 30 *Salmonella* Enteritidis PT 30 isolates recovered between 2001 and 2006 from a 2001 outbreak-associated almond orchard were included in the analysis. The CFSAN SNP Pipeline was selected to measure the genetic distances among the isolates. WGS was used to predict antimicrobial resistance (AMR), virulence genes, and presence of plasmid DNA.

## Materials and methods

### Isolate selection

Isolates were retrieved from enrichment of raw almond kernels and inshell almonds (survey isolates), and from swabs that were dragged across the floor of one of the 2001 outbreak-associated orchards (orchard isolates). A total of 15,505 ∼1-kg samples from different lots of raw almond kernels (14,949) and inshell almonds (455) were collected upon receipt at several almond processors located throughout California from the 2001–2005 [5], 2006–2007 [6], 2010 ([15]; present study), and 2013 ([7]; present study) harvests, as described previously. All samples were coded to keep the identities of the processors confidential; the geographic origins of the samples were unknown. The Safe Food Alliance, Kingsburg, CA (formerly American Council for Food Safety and Quality, Fresno, CA) analyzed a 100-g subsample from each lot of almonds by enriching for the presence of *Salmonella* [5]. In addition, the MPN of *Salmonella* was determined for 118 samples by enriching one or more additional samples weighing from ∼0.25 to 100 g. *Salmonella* isolates were stored at −80°C.

A total of 171 *Salmonella* survey isolates were selected for the present study: for each individual positive almond lot, a single *Salmonella* isolate retrieved from the initial 100-g subsample (153 positive samples, 148 isolates; three, one, and one isolate from 2001, 2002, and 2003, respectively, were lost), any isolates recovered from MPN or secondary enrichments that differed from the initial serotype (12), and unique isolates recovered from secondary enrichments of initially negative samples (11) (Tables 1 and S1 in S1 File). Traditional serotyping was done for each banked *Salmonella* by the California Animal Health and Food Safety Laboratory System (Davis, CA), and phage typing for *Salmonella* Enteritidis isolates was done by the National Veterinary Services Laboratory (Ames, IA) (Table 1). *Salmonella* Enteritidis PT 9c LJH1024 (obtained from Robert Mandrell, U.S. Department of Agriculture, Agricultural Research Service; RM4635 or G04-101 [8], raw almond isolate from the 2004 outbreak) and *Salmonella* Enteritidis PT 30 LJH0608 (raw almond isolate from the 2001 outbreak that was deposited to the American Type Culture Collection as ATCC BAA-1045) were also included in some of the analyses (S2 Table in S1 File).

Several 2001 outbreak-associated almond orchards sampled by investigators during the 2001 outbreak investigation were positive for *Salmonella* Enteritidis PT 30 [1]. One of these orchards was sampled every year from 2001 [1] through 2006 [2]. Briefly, sterile gauze swabs

**Table 1. Results of the WGS serotyping and number of plasmids for *Salmonella* survey isolates retrieved from raw almonds.**

| Strain designation | Isolation year | Accession number | WGS serotyping | | Traditional serotyping | Number of plasmids |
|---|---|---|---|---|---|---|
| | | | Subspecies | WGS serotyping | | |
| LJH0651 | 2001 | SRR23719048 | enterica | Brandenburg | Brandenburg | 0 |
| LJH0652 | 2001 | SRR23719047 | enterica | Thompson | Thompson | 0 |
| LJH0653 | 2001 | SRR23718942 | enterica | Montevideo | Montevideo | 4 |
| LJH0654 | 2001 | SRR23718931 | enterica | Montevideo | Montevideo | 0 |
| LJH0655 | 2001 | SRR23718920 | enterica | Brandenburg | Brandenburg | 0 |
| LJH0656 | 2001 | SRR23718909 | enterica | Newport | Newport | 2 |
| LJH0657 | 2001 | SRR23718898 | enterica | Montevideo | Montevideo | 0 |
| LJH0658 | 2001 | SRR23718887 | enterica | Give | Nancy (Nchanga) | 3 |
| LJH0659 | 2001 | SRR23719007 | enterica | Montevideo | Montevideo | 1 |
| LJH0666 | 2002 | SRR23718996 | enterica | Typhimurium | Typhimurium | 0 |
| LJH0667 | 2002 | SRR23719046 | enterica | Senftenberg | Senftenberg | 4 |
| LJH0668 | 2002 | SRR23718976 | enterica | Give | Give | 4 |
| LJH0669 | 2002 | SRR23718965 | enterica | Typhimurium | Typhimurium | 0 |
| LJH0687 | 2002 | SRR23718954 | enterica | 1,4,[5],12:i:- | 1,4,[5],12:i:- | 1 |
| LJH0690 | 2002 | SRR23719038 | enterica | Oranienburg | Oranienburg | 1 |
| LJH0691 | 2002 | SRR23719027 | enterica | Oranienburg | Oranienburg | 1 |
| LJH0692 | 2002 | SRR23719016 | enterica | Worthington | Worthington | 1 |
| LJH0693 | 2002 | SRR23718946 | enterica | Heidelberg | Worthington | 2 |
| LJH0694 | 2002 | SRR23718944 | enterica | Oranienburg | Muenchen | 1 |
| LJH0695 | 2002 | SRR23718943 | enterica | Newport | Heidelberg | 1 |
| LJH0696 | 2002 | SRR23718941 | enterica | Thompson | Agona | 0 |
| LJH0697 | 2002 | SRR23718940 | enterica | Newport | Newport | 1 |
| LJH0698 | 2002 | SRR23718939 | enterica | Manhattan | Agona | 1 |
| LJH0713 | 2002 | SRR23718938 | enterica | Senftenberg | Senftenberg | 0 |
| LJH0714 | 2002 | SRR23718937 | enterica | Lomalinda | Lomalinda | 1 |
| LJH0715 | 2002 | SRR23718936 | enterica | Tennessee | Tennessee | 1 |
| LJH0716 | 2002 | SRR23718935 | enterica | Braenderup | Braenderup | 2 |
| LJH0717 | 2002 | SRR23718934 | enterica | Typhimurium | Typhimurium | 0 |
| LJH0718 | 2002 | SRR23718933 | enterica | Schwarzengrund | Schwarzengrund | 3 |
| LJH0719 | 2002 | SRR23718932 | enterica | Montevideo | Montevideo | 1 |
| LJH0720 | 2002 | SRR23718930 | enterica | Anatum | Anatum | 1 |
| LJH0721 | 2002 | SRR23718929 | enterica | Tennessee | Tennessee | 0 |
| LJH0722 | 2002 | SRR23718928 | enterica | Infantis | Infantis | 0 |
| LJH0724 | 2002 | SRR23718927 | enterica | Zerifin | Zerifin | 4 |
| LJH0725 | 2003 | SRR23718926 | enterica | Horsham | Brandenburg | 0 |
| LJH0726 | 2003 | SRR23718925 | enterica | Indiana | Indiana | 7 |
| LJH0738 | 2003 | SRR23718924 | enterica | 1,4,[5],12:i:- | Typhimurium | 1 |
| LJH0739 | 2003 | SRR23718923 | enterica | Thompson | Thompson | 0 |
| LJH0740 | 2003 | SRR23718922 | enterica | Thompson | Thompson | 1 |
| LJH0741 | 2003 | SRR23718921 | enterica | Thompson | Thompson | 1 |
| LJH0742 | 2003 | SRR23718919 | enterica | Sandiego | Sandiego | 1 |
| LJH0751 | 2003 | SRR23718918 | enterica | Newport | Newport | 1 |
| LJH0752 | 2003 | SRR23718917 | enterica | Oranienburg | Othmarschen | 1 |
| LJH0753 | 2003 | SRR23718916 | enterica | Istanbul | Istanbul | 3 |
| LJH0754 | 2003 | SRR23718915 | enterica | Muenchen | Newport | 1 |
| LJH0759 | 2003 | SRR23718914 | enterica | Montevideo | Montevideo | 0 |

*(Continued)*

**Table 1.** (Continued)

| Strain designation | Isolation year | Accession number | WGS serotyping | | Traditional serotyping | Number of plasmids |
|---|---|---|---|---|---|---|
| | | | Subspecies | WGS serotyping | | |
| LJH0760 | 2003 | SRR23718913 | *enterica* | Montevideo | Montevideo | 1 |
| LJH0761 | 2003 | SRR23718912 | *enterica* | Typhimurium | Typhimurium | 1 |
| LJH0762 | 2003 | SRR23718911 | *enterica* | Enteritidis | Enteritidis | 1 |
| LJH0783 | 2004 | SRR23718910 | *enterica* | Liverpool | Liverpool | 0 |
| LJH0784 | 2004 | SRR23718908 | *enterica* | Braenderup | Braenderup | 1 |
| LJH0787 | 2004 | SRR23718907 | *enterica* | Anatum | Anatum | 3 |
| LJH0788 | 2004 | SRR23718906 | *enterica* | Typhimurium | Typhimurium var. Copenhagen | 1 |
| LJH0789 | 2004 | SRR23718905 | *enterica* | Montevideo | Montevideo | 0 |
| LJH0790 | 2004 | SRR23718904 | *enterica* | Horsham | Horsham | 0 |
| LJH0791 | 2004 | SRR23718903 | *enterica* | Thompson | Thompson | 1 |
| LJH0792 | 2004 | SRR23718902 | *enterica* | Thompson | Thompson | 1 |
| LJH0793 | 2004 | SRR23718901 | *enterica* | Thompson | Thompson | 1 |
| LJH0794 | 2004 | SRR23718900 | *enterica* | Thompson | Thompson | 1 |
| LJH1011 | 2004 | SRR23718899 | *enterica* | Senftenberg | Senftenberg | 0 |
| LJH1012 | 2004 | SRR23718897 | *enterica* | Anatum | Anatum | 2 |
| LJH1013 | 2005 | SRR23718896 | *enterica* | Newport | Saintpaul | 0 |
| LJH1019 | 2005 | SRR23718895 | *enterica* | Give | Give | 2 |
| LJH1020 | 2005 | SRR23718894 | *enterica* | Montevideo | Montevideo | 1 |
| LJH1021 | 2005 | SRR23718893 | *enterica* | Heidelberg | Heidelberg | 2 |
| LJH1022 | 2005 | SRR23718892 | *enterica* | Mbandaka | Mbandaka | 0 |
| LJH1023 | 2005 | SRR23718891 | *enterica* | Enteritidis | Enteritidis | 3 |
| LJH1025 | 2005 | SRR23718890 | *enterica* | Typhimurium | Typhimurium | 0 |
| LJH1026 | 2005 | SRR23718889 | *enterica* | Manhattan | Untypeable | 0 |
| LJH1027 | 2005 | SRR23718888 | *enterica* | Muenchen | Muenchen | 0 |
| LJH1028 | 2005 | SRR23718886 | *enterica* | Enteritidis | Enteritidis | 1 |
| LJH1029 | 2005 | SRR23718885 | *enterica* | Enteritidis | Untypeable | 1 |
| LJH1030 | 2005 | SRR23718884 | *enterica* | Tennessee | Tennessee | 0 |
| LJH1043 | 2005 | SRR23718883 | *enterica* | Typhimurium | Typhimurium var. Copenhagen | 1 |
| LJH1044 | 2005 | SRR23718882 | *enterica* | Kentucky | Kentucky | 2 |
| LJH1045 | 2005 | SRR23718881 | *enterica* | Montevideo | Montevideo | 0 |
| LJH1046 | 2005 | SRR23718880 | *enterica* | Enteritidis | Enteritidis | 1 |
| LJH1047 | 2005 | SRR23718879 | *enterica* | Enteritidis | Enteritidis | 0 |
| LJH1048 | 2005 | SRR23718878 | *enterica* | Enteritidis | Enteritidis | 1 |
| LJH1049 | 2005 | SRR23719008 | *enterica* | Enteritidis | Enteritidis | 1 |
| LJH1052 | 2005 | SRR23719006 | *enterica* | Duisburg | Duisburg | 3 |
| LJH1054 | 2006 | SRR23719005 | *enterica* | Typhimurium | Typhimurium | 1 |
| LJH1055 | 2006 | SRR23719004 | *enterica* | Typhimurium | Typhimurium | 1 |
| LJH1056 | 2006 | SRR23719003 | *enterica* | Typhimurium | Typhimurium | 1 |
| LJH1058 | 2006 | SRR23719002 | *enterica* | Muenchen | Muenchen | 0 |
| LJH1059 | 2006 | SRR23719001 | *enterica* | Enteritidis | Enteritidis | 1 |
| LJH1063 | 2006 | SRR23719000 | *enterica* | Anatum | Anatum | 1 |
| LJH1067 | 2006 | SRR23718999 | *diarizonae* (IIIb) | | III 50:k:- | 0 |
| LJH1068 | 2006 | SRR23718998 | *enterica* | Newport | Newport | 0 |
| LJH1070 | 2006 | SRR23718997 | *enterica* | Heidelberg | Heidelberg | 2 |
| LJH1071 | 2006 | SRR23718995 | *enterica* | Give | Give | 2 |
| LJH1076 | 2006 | SRR23718994 | *enterica* | Muenchen | Muenchen | 0 |

*(Continued)*

**Table 1.** (Continued)

| Strain designation | Isolation year | Accession number | WGS serotyping | | Traditional serotyping | Number of plasmids |
|---|---|---|---|---|---|---|
| | | | Subspecies | WGS serotyping | | |
| LJH1080 | 2006 | SRR23718993 | *enterica* | Muenchen | Muenchen | 0 |
| LJH1082 | 2006 | SRR23718992 | *enterica* | 1,4,[5],12:i:- | 1,4,[5],12:i:- | 1 |
| LJH1083 | 2006 | SRR23718991 | *enterica* | Muenchen | Muenchen | 0 |
| LJH1084 | 2006 | SRR23718990 | *enterica* | Newport | Newport | 1 |
| LJH1085 | 2006 | SRR23718989 | *enterica* | Newport | Muenchen | 0 |
| LJH1087 | 2006 | SRR23718988 | *enterica* | Muenchen | Muenchen | 0 |
| LJH1088 | 2006 | SRR23718987 | *enterica* | Muenchen | Newport | 0 |
| LJH1089 | 2006 | SRR23718985 | *enterica* | Horsham | Horsham | 0 |
| LJH1090 | 2006 | SRR23719045 | *enterica* | Muenchen | Muenchen | 0 |
| LJH1094 | 2006 | SRR23718986 | *enterica* | Montevideo | Montevideo | 0 |
| LJH1095 | 2006 | SRR23718984 | *enterica* | 1,4,[5],12:i:- | 1,4,[5],12:i:- | 1 |
| LJH1096 | 2006 | SRR23718983 | *enterica* | Enteritidis | Enteritidis | 2 |
| LJH1097 | 2006 | SRR23718982 | *enterica* | Oranienburg | Oranienburg | 1 |
| LJH1098 | 2006 | SRR23718981 | *enterica* | 1,4,[5],12:i:- | 1,4,[5],12:i:- | 1 |
| LJH1099 | 2006 | SRR23718980 | *enterica* | Meleagridis | Meleagridis | 0 |
| LJH1100 | 2006 | SRR23718979 | *enterica* | Agona | Agona | 1 |
| LJH1101 | 2006 | SRR23718978 | *enterica* | Muenchen | Muenchen | 0 |
| LJH1102 | 2006 | SRR23718977 | *enterica* | Montevideo | Montevideo | 0 |
| LJH1103 | 2006 | SRR23718975 | *enterica* | Enteritidis | Enteritidis | 1 |
| LJH1104 | 2006 | SRR23718974 | *enterica* | Enteritidis | Enteritidis | 1 |
| LJH1105 | 2006 | SRR23718973 | *enterica* | Give | Give | 1 |
| LJH1106 | 2006 | SRR23718972 | *enterica* | Agona | Agona | 2 |
| LJH1107 | 2006 | SRR23718971 | *enterica* | Newport | Newport | 2 |
| LJH1108 | 2006 | SRR23718970 | *enterica* | Muenchen | Muenchen | 0 |
| LJH1109 | 2006 | SRR23718969 | *enterica* | Enteritidis | Enteritidis | 4 |
| LJH1133 | 2007 | SRR23718968 | *enterica* | Newport | Newport | 0 |
| LJH1134 | 2007 | SRR23718967 | *enterica* | Cerro | Cerro | 0 |
| LJH1135 | 2007 | SRR23718966 | *enterica* | Muenchen | Muenchen | 0 |
| LJH1136 | 2007 | SRR23718964 | *enterica* | Cerro | Cerro | 0 |
| LJH1137 | 2007 | SRR23718963 | *enterica* | Manhattan | Manhattan | 1 |
| LJH1138 | 2007 | SRR23718962 | *enterica* | Newport | Newport | 2 |
| LJH1139 | 2007 | SRR23718961 | *enterica* | Thompson | Thompson | 0 |
| LJH1140 | 2007 | SRR23718960 | *enterica* | Irumu | Irumu | 0 |
| LJH1141 | 2007 | SRR23718959 | *enterica* | Typhimurium | Typhimurium | 2 |
| LJH1142 | 2007 | SRR23718958 | *arizonae* (IIIa) | | IIIa 18:z32:- | 0 |
| LJH1143 | 2007 | SRR23718957 | *enterica* | Oranienburg | Othmarschen | 1 |
| LJH1144 | 2007 | SRR23718956 | *enterica* | Typhimurium | I 4,12:i:- | 3 |
| LJH1145 | 2007 | SRR23718955 | *enterica* | Brandenburg | Brandenburg | 2 |
| LJH1146 | 2007 | SRR23718953 | *enterica* | Thompson | Thompson | 0 |
| LJH1147 | 2007 | SRR23718952 | *enterica* | Give | Bredeney | 1 |
| LJH1148 | 2007 | SRR23718951 | *enterica* | Newport | Newport | 0 |
| LJH1149 | 2007 | SRR23718950 | *enterica* | Cerro | Cerro | 0 |
| LJH1150 | 2007 | SRR23719044 | *enterica* | Senftenberg | Senftenberg | 0 |
| LJH1151 | 2007 | SRR23719043 | *enterica* | Muenchen | Untypeable | 1 |
| LJH1154 | 2007 | SRR23719042 | *enterica* | Senftenberg | Senftenberg | 0 |
| LJH1248-1 | 2010 | SRR23719041 | *enterica* | Newport | Newport | 1 |

*(Continued)*

**Table 1.** (Continued)

| Strain designation | Isolation year | Accession number | WGS serotyping | | Traditional serotyping | Number of plasmids |
|---|---|---|---|---|---|---|
| | | | Subspecies | WGS serotyping | | |
| LJH1249-1 | 2010 | SRR23719040 | *enterica* | Give | Give | 1 |
| LJH1250-1 | 2010 | SRR23719039 | *enterica* | Infantis | Infantis | 0 |
| LJH1251-1 | 2010 | SRR23719037 | *enterica* | Give | Give | 1 |
| LJH1252-1 | 2010 | SRR23719036 | *enterica* | Infantis | Infantis | 0 |
| LJH1266-1 | 2010 | SRR23719035 | *arizonae* (IIIa) | | II:17:g,t:- | 0 |
| LJH1267-1 | 2010 | SRR23719034 | *enterica* | Mbandaka | Mbandaka | 0 |
| LJH1268-1 | 2010 | SRR23719033 | *enterica* | Infantis | Infantis | 0 |
| LJH1269-1 | 2010 | SRR23719032 | *enterica* | Duisburg | Duisburg | 1 |
| LJH1270-1 | 2010 | SRR23719031 | *enterica* | Heidelberg | Heidelberg | 2 |
| LJH1271-1 | 2010 | SRR23719030 | *enterica* | Infantis | Infantis | 0 |
| LJH1272-1 | 2010 | SRR23719029 | *enterica* | Enteritidis | Enteritidis | 1 |
| LJH1273-1 | 2010 | SRR23719028 | *enterica* | Newport | Newport | 1 |
| LJH1276-1 | 2010 | SRR23719026 | *enterica* | Oranienburg | Othmarschen | 1 |
| LJH1277-1 | 2010 | SRR23719025 | *enterica* | Heidelberg | Heidelberg | 2 |
| LJH1278-1 | 2010 | SRR23719024 | *enterica* | Newport | Newport | 1 |
| LJH1618-1 | 2013 | SRR23719023 | *enterica* | 1,4,[5],12:i:- | 1,4,[5],12:i:- | 1 |
| LJH1619-1 | 2013 | SRR23719022 | *enterica* | Muenchen | I 6:8:d:z6 | 1 |
| LJH1620-1 | 2013 | SRR23719021 | *enterica* | Muenchen | Untypeable | 1 |
| LJH1622-1 | 2013 | SRR23719020 | *enterica* | Heidelberg | Heidelberg | 1 |
| LJH1623-1 | 2013 | SRR23719019 | *enterica* | Montevideo | Montevideo | 0 |
| LJH1624-1 | 2013 | SRR23719018 | *diarizonae* (IIIb) | P:k:z35 | Untypeable | 0 |
| LJH1628-1 | 2013 | SRR23719017 | *enterica* | Montevideo | Montevideo | 1 |
| LJH1629-1 | 2013 | SRR23719015 | *enterica* | Give | Give | 1 |
| LJH1630-1 | 2013 | SRR23719014 | *enterica* | Cerro | Cerro | 2 |
| LJH1631-1 | 2013 | SRR23719013 | *enterica* | Give | Give | 2 |
| LJH1633-1 | 2013 | SRR23719012 | *enterica* | Enteritidis | Enteritidis | 2 |
| LJH1660-1 | 2013 | SRR23719011 | *arizonae* (IIIa) | | IIIa 41:z23: - | 0 |
| LJH1661-1 | 2013 | SRR23719010 | *enterica* | Muenchen | Muenchen | 0 |
| LJH1662-1 | 2013 | SRR23719009 | *enterica* | 1,4,[5],12:i:- | Typhimurium | 1 |
| LJH1664-1 | 2013 | SRR23718949 | *diarizonae* (IIIb) | | IIIb 50:r:z | 0 |
| LJH1665 | 2013 | SRR23718948 | *enterica* | 1,4,[5],12:i:- | 1,4,[5],12:i:- | 1 |
| LJH1673 | 2013 | SRR23718947 | *enterica* | Enteritidis | Enteritidis | 2 |
| LJH1676 | 2013 | SRR23718945 | *enterica* | 1,4,[5],12:i:- | Typhimurium | 1 |

attached to a string and soaked in full-strength evaporated skim milk were pulled along the orchard floor in a standardized manner. Four individual swabs were pooled and a procedure designed for recovering *Salmonella* from poultry houses was used to enrich the samples. Three of six (50%) pooled swab samples collected in 2001 and 53 of 228 (23%) samples collected between 2002 and 2006 were positive for *Salmonella*; every isolate was identified as *Salmonella* Enteritidis PT 30. A total of 30 *Salmonella* Enteritidis PT 30 orchard isolates from 2001 (3), 2002 (12), 2003 (10), 2005 (2), and 2006 (3) were analyzed in the present study (S2 Table in S1 File). One orchard isolate from 2002 and all orchard isolates from 2004 (25) were not available and thus not included.

Using the pathogen detection tool associated with the NCBI database (https://www.ncbi. nlm.nih.gov/pathogens), the sequence read archive (SRA) data were downloaded for 12

*Salmonella* Enteritidis PT 30 clinical isolates from the 2001 almond outbreak and for four clinical isolates from the 2006 almond outbreak (S3 Table in S1 File).

## Whole genome sequencing

Isolates were retrieved from frozen glycerol stock and plated on tryptic soy agar (TSA). Following overnight incubation at 37˚C, one colony was inoculated into 2 ml of tryptic soy broth (TSB) and incubated for 24 h at 37˚C, with shaking at 120 rpm, before being pelleted by centrifugation at $14,000 \times g$ for 2 min. Genomic DNA, for the isolates in S2 Table (S1 File), was extracted with the QIAamp DNA minikit (Qiagen, Valencia, CA) following the manufacturer's directions. The 150-bp paired-end libraries were constructed for each purified DNA with the Illumina Nextera DNA flex library kit following the manufacturer's directions (Illumina Inc., San Diego, CA). Pooled samples were sequenced on an Illumina 4000 HiSeq system by the DNA Technologies and Expression Analysis Core at the UC Davis Genome Center. DNA, for all the survey isolates, was sequenced as described by Emond-Rheault et al. in 2020 [16]. All sequence data obtained in this study were deposited to the NCBI pathogen database under the BioProject accession number PRJNA941918 (Tables 1 and S2 in S1 File) and PRJNA951760 (S2 Table in S1 File).

## Quality control and genome assembly

Raw read quality was assessed with FastQC (v0.11.8) (https://www.bioinformatics.babraham. ac.uk/projects/fastqc/). Illumina adapter sequences and low-quality sequences were trimmed using Trimmomatic version 0.36 [17]. Reads were assembled de novo with SPAdes version v3.13.0 Genome Assembler [18]. Draft *Salmonella* genome assemblies were serotyped with SeqSero [19] and by aligning with BLASTn against the nonredundant nucleotide sequence database.

## Core genome SNP typing

De novo genomes were used to build core genome single nucleotide trees using the Parsnp aligner v1.2 with default parameters and the requirement for all genomes to be included in the analysis [20]. The 171 *Salmonella* genomes from the survey were mapped to the complete reference genome *Salmonella* Typhimurium str. LT2 (NCBI accession: AE006468), resulting in an alignment of 50% of the core genome. The whole-genome phylogeny was constructed with FastTree2 [21]. iTOL (http://itol.embl.de/) was used to visualize the tree and annotate it with the AMR genes [22].

## Genetic distance

Genetic distance between multiple isolates of the same serovar was evaluated as the number of SNP differences detected with the CFSAN SNP Pipeline [12]. The CFSAN SNP Pipeline v2.0.2 was installed on a local ubuntu platform with all the executable software dependencies. Prior to analyzing our data, we used the data set provided for testing the reproducibility of the software to confirm that our installation of the CFSAN SNP Pipeline was correct. Reference-based alignments were created for a set of samples and used to generate the SNP matrix. Because the software was developed for closely related genome sequences, where available, complete assembled reference genomes were downloaded from NCBI (20; S4 Table in S1 File). The phylogenies were inferred with MEGA7 [23] using the neighbor-joining method [24] based on the obtained SNP matrix formatted as a FASTA file generated by the CFSAN SNP pipeline v2.0.2.

## Virulence and antibiotic resistance (AMR) gene prediction

The presence of resistance genes, as well as point mutations, were determined using ResFinder 4.1 (Center for Genomic Epidemiology, https://cge.food.dtu.dk/services/ResFinder) with a setting threshold of 90% and minimum length of 60% [25–27]. The assembled draft genomes for the survey isolates (171) were used as an input to identify *Salmonella* AMR genes associated with resistance to aminoglycoside, β-lactam, chloramphenicol, colistin, fluoroquinolone, fosfomycin, glycopeptide, macrolide, sulfonamide, tetracycline, and trimethoprim antibiotics. The VF analyzer pipeline was used to screen the assembled draft genomes against the Virulence Factor Database (VFDB) for potential virulence factors [28].

## Plasmid detection and reconstruction

Plasmids from genome assemblies were typed and reconstructed using MOB-suite v3.1.0 with the default parameters [29, 30]. To determine the AMR and virulence genes that were plasmid-borne, the reconstructed plasmids were screened against the CARD (https://card.mcmaster.ca) and VFDB [28] databases, respectively, using Abricate v0.5 (https://github.com/tseemann/abricate) with the same parameters as described above.

## Results

### Phylogenetic analysis of *Salmonella* isolates retrieved from raw almonds

Genomes were initially compared with Parsnp because the CFSAN SNP Pipeline is not recommended for relatively distant bacteria (greater than a few hundred SNP differences). A maximum-likelihood phylogeny tree was constructed with Parsnp based on alignment of the 171 *Salmonella* assembled genomes [20]. All the isolates belonged to the species *enterica*, with most (165) belonging to the subspecies *enterica* (Fig 1 and Table 1). Three isolates were classified as subspecies *diarizonae* and three as subspecies *arizonae*. The phylogenetic tree clustered the isolates by serovar (Fig 1). The serovars identified by classical serotyping matched the serovars predicted by WGS for 92% of the isolates (Table 1). However, 14 isolates clustered with serotypes that differed from those to which they were initially assigned by traditional serotyping (Table 1). To eliminate the possibility of manipulation errors, these isolates were resequenced and their serotype was confirmed with SeqSero and with BLASTn against the nonredundant nucleotide sequence database. Results were consistent with the initial WGS serovar prediction.

Based on the core genome SNP typing, a total of 32 unique *Salmonella* serovars were identified. Of these, 22 *Salmonella* serovars were isolated two or more times between 2001 and 2013: Enteritidis ($n = 16$), Muenchen ($n = 16$), Newport ($n = 15$), Montevideo ($n = 14$), Typhimurium ($n = 12$), Thompson ($n = 11$), Give ($n = 10$), 1,4,[5],12:i- ($n = 9$), Oranienburg ($n = 7$), Heidelberg ($n = 6$), Infantis ($n = 5$), Senftenberg ($n = 5$), Anatum ($n = 4$), Cerro ($n = 4$), Brandenburg ($n = 3$), Duisburg ($n = 3$), Horsham ($n = 3$), Manhattan ($n = 3$), Tennessee ($n = 3$), Agona ($n = 2$), Braenderup ($n = 2$), and Mbandaka ($n = 2$) (Fig 1 and Table 1). Three isolates initially identified as *Salmonella* Othmarschen (LJH0752, LJH1143, and LJH1276-1) clustered with *Salmonella* Oranienburg. Both serotypes have a similar antigenic formula, 6,7,14:m,t:-, which makes them difficult to distinguish by serological methods [31]. Serovar 1,4,[5],12:i- is a monophasic variant of *Salmonella* Typhimurium, and three of 14 isolates initially identified as Typhimurium (LJH0738, LJH1662-1, LJH1676) clustered with 1,4,[5],12:i:-. One *Salmonella* initially identified as 1,4,[5],12:i:- (LJH1144) clustered with Typhimurium.

Among the five isolates that were untypeable by classical serotyping, two clustered with serovar Muenchen (LJH1151, LJH1620-1), one with Enteritidis (LJH1029), and one with

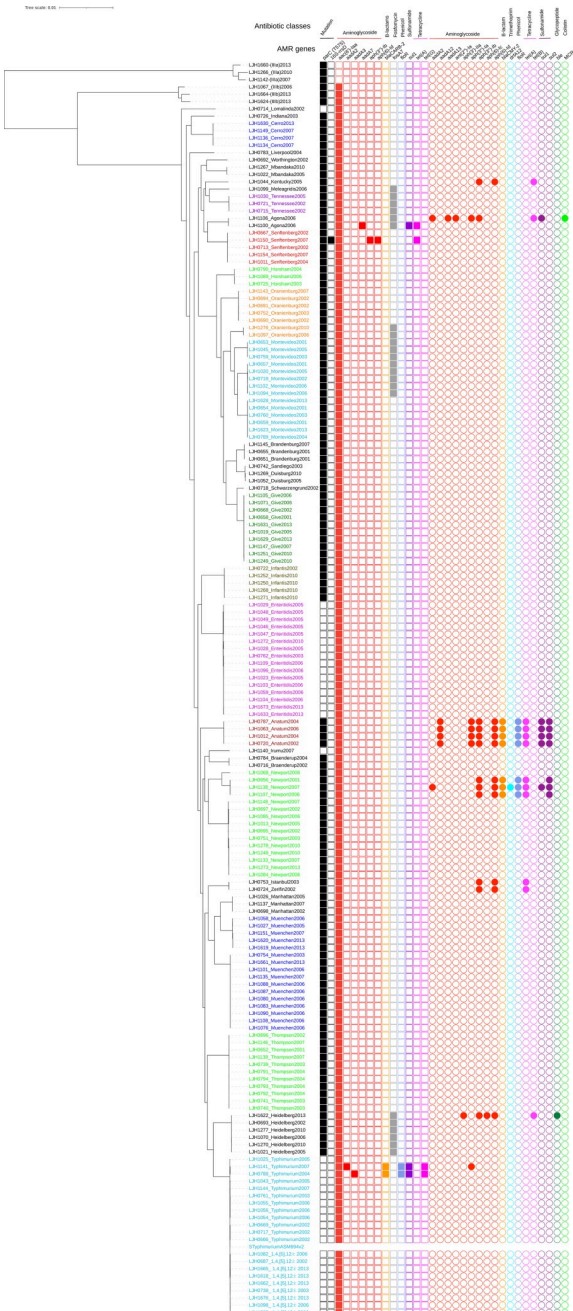

**Fig 1. Maximum likelihood tree based on SNPs identified by aligning 171 *de novo* assemblies to the reference chromosome of *Salmonella* Typhimurium LT2 ASM694v2 with Parsnp and genotypic antimicrobial resistance (AMR).** The scale bar is in number of SNPs. The colors in the phylogenetic tree represent different serogroups; squares represent a chromosomal location and circles represent a plasmid location for AMR genes. AMR genes are color-coded by antibiotic classes.

Manhattan (LJH1026). A single untypeable isolate (LJH1624-1) clustered with *Salmonella diarizonae* (LJH1664) and was identified as *Salmonella diarizonae* with BLASTn and SeqSero [19]. Fifteen unique serotypes, predicted with BLASTn and SeqSero, matched the serological serotyping. It was difficult to predict some serovars due to the limited number of

representative genomes in the SeqSero database. *Salmonella* Zerifin LJH0724 was identified as *Salmonella* Istanbul with SeqSero and clustered with the Istanbul isolates in the Parsnp phylogeny tree (Fig 1). Conflicting serotyping results between traditional methods and WGS have been reported in previous studies [32–35]. Serotyping based on WGS has been increasingly used by public health laboratories and federal agencies to replace the current standard of phenotypic serotyping [10, 32, 34, 36]. In the present study, serotype was assigned based on the WGS analysis when the serovar assignment was identical among the phylogenetic relationship, SeqSero, and BLASTn analyses. Except for a single strain of *Salmonella* Zerifin, all isolates (170 of 171) were assigned a serotype based on the WGS analysis.

## Genetic distance within each serovar

The CFSAN SNP Pipeline was used to evaluate the genetic distance among multiple isolates within each serotype. *Salmonella* Enteritidis isolates clustered in three groups by phage type: PT 8, PT 9c, and PT 30 (Fig 2). All five *Salmonella* Enteritidis PT 8 isolates were retrieved in 2005 from separate almond lots and had less than three SNP differences, classifying them as clonal isolates. *Salmonella* Enteritidis PT 9c isolated in 2005 (LJH1028) and 2010 (LJH1272), differed by 5 and 13 SNPs, respectively, from a 2004 *Salmonella* Enteritidis PT 9c outbreak isolate (LJH1024; Fig 2).

The genomes of *Salmonella* Enteritidis PT 30 recovered from survey almonds (eight isolates (Table 1): LJH0762 [2003], LJH1023 [2005], LJH1104 [2006], LJH1059 [2006], LJH1096 [2006], LJH1109 [2006], LJH1633 [2013], LJH1673 [2013]), the 2001 outbreak-associated orchard (30 isolates; S2 Table in S1 File), and a 2001 outbreak-associated almond isolate (LJH0608) were compared to *Salmonella* Enteritidis PT 30 genomes of clinical isolates from almond outbreaks in 2001 (12 isolates) and 2006 (four isolates) (S3 Table in S1 File).

*Salmonella* Enteritidis PT 30 isolates formed two clusters (Fig 2). One consisted of a single survey isolate (LJH0762), recovered in 2003, that differed from LJH0608 by 48 SNPs (Fig 2 and S5 Table in S1 File). All other survey and clinical isolates (*n* = 38) clustered in a single group with LJH0608 that differed from each other by ≤18 SNPs (Fig 2) indicating that the isolates are from a common origin. Almond isolates from 2001 to 2013 had 2 to 13 SNP differences compared with the 2001 outbreak-associated almond isolate *Salmonella* Enteritidis PT 30 LJH0608 (Fig 2 and S5 Table in S1 File). Although this isolate was recovered from recalled almonds in 2001, the almonds were harvested in the fall of 2000 [1], a span of 14 years (2000–2013). The orchard isolates from 2001 to 2006 differed by 0 to 12 SNPs within their genomes and by 3 to 13 SNPs with the clinical genomes. The SNP differences ranged from zero to eight within the 12 clinical isolates from the 2001 outbreak and from one to 13 within the four clinical isolates from the 2006 outbreak. Among the clinical isolates from 2001 and 2006, the SNP differences ranged from four to 13, indicating that the isolates are from a common origin.

Almond, orchard, and clinical isolates of *Salmonella* Enteritidis PT 30 isolated from 2001 through 2013 are closely related strains. The persistence of *Salmonella* Enteritidis PT 30 in an almond orchard over 6 years was reported previously [2]. The SNP analysis confirmed the PFGE results obtained for these isolates. Almonds from the 2001 outbreak-associated orchards were purposefully excluded from the raw almond survey. Because survey samples were coded and the sources unknown, it is possible that samples harvested from outbreak-associated orchards were inadvertently included. It is also possible that survey almonds were cross contaminated with almonds harvested from outbreak-associated orchards via harvest equipment or at a common almond huller-sheller, or that *Salmonella* Enteritidis PT 30 was spread over a broader geographic region than recognized as associated with the 2001 outbreak.

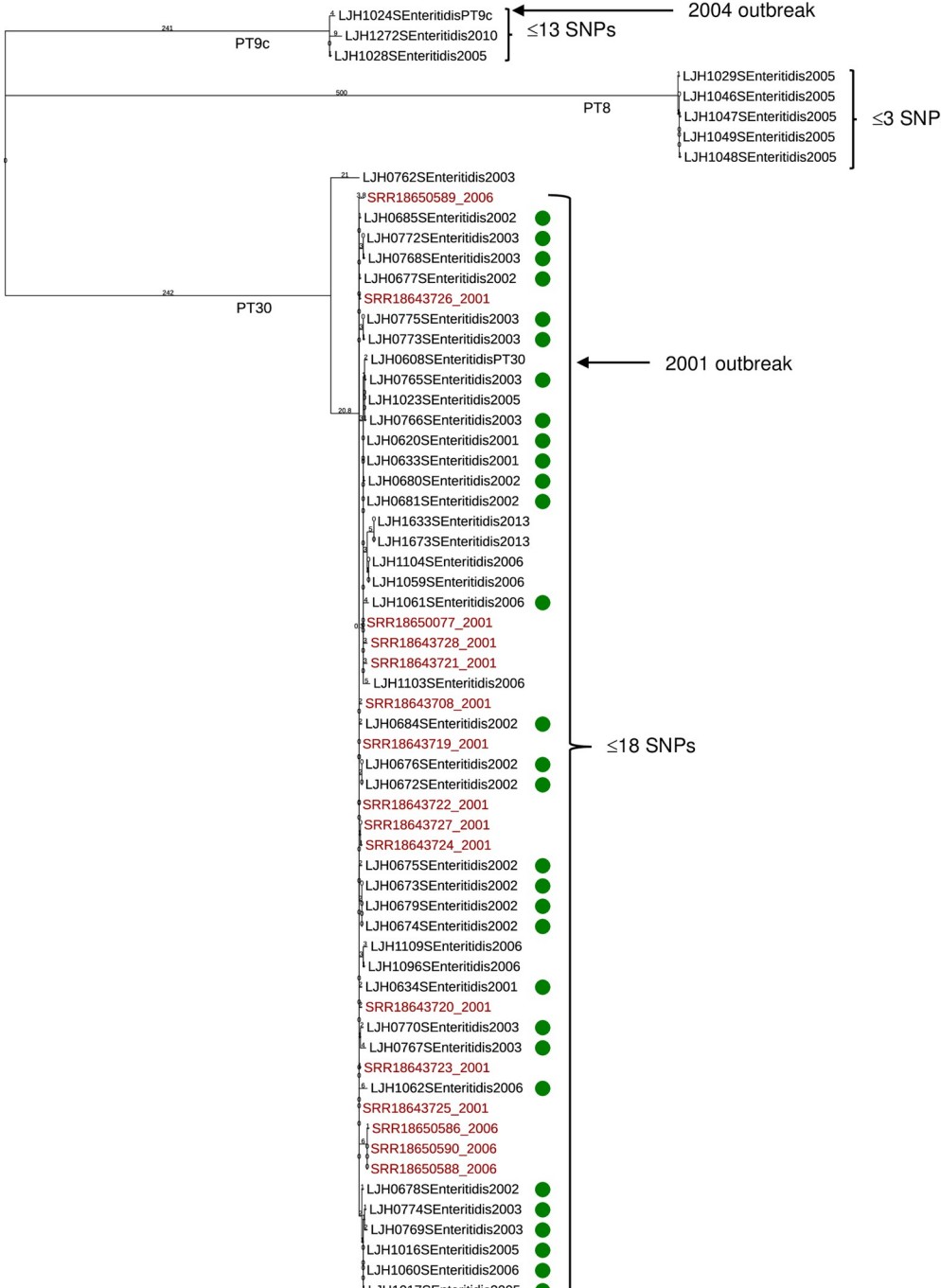

**Fig 2. Phylogenetic tree of *Salmonella* Enteritidis generated with the CFSAN SNP Pipeline.** The branch lengths represent the SNP distances among the isolates. Almond survey isolates (black text) and outbreak-associated orchard isolates (green dot) were compared to clinical isolates (red text) from the 2001 and 2006 outbreaks (retrieved from the NCBI database). Almond isolates of *Salmonella* Enteritidis PT 9c (LJH1024) and *Salmonella* Enteritidis PT 30 (LJH0608) from the 2004 and 2001 outbreaks, respectively, were included for comparison.

*Salmonella* Enteritidis was not the only serovar for which clonal isolates were recovered from almonds in different years. *Salmonella* Montevideo survey isolates clustered into three groups, with more than 100 SNPs between them (Fig 3). Within each of these clusters there

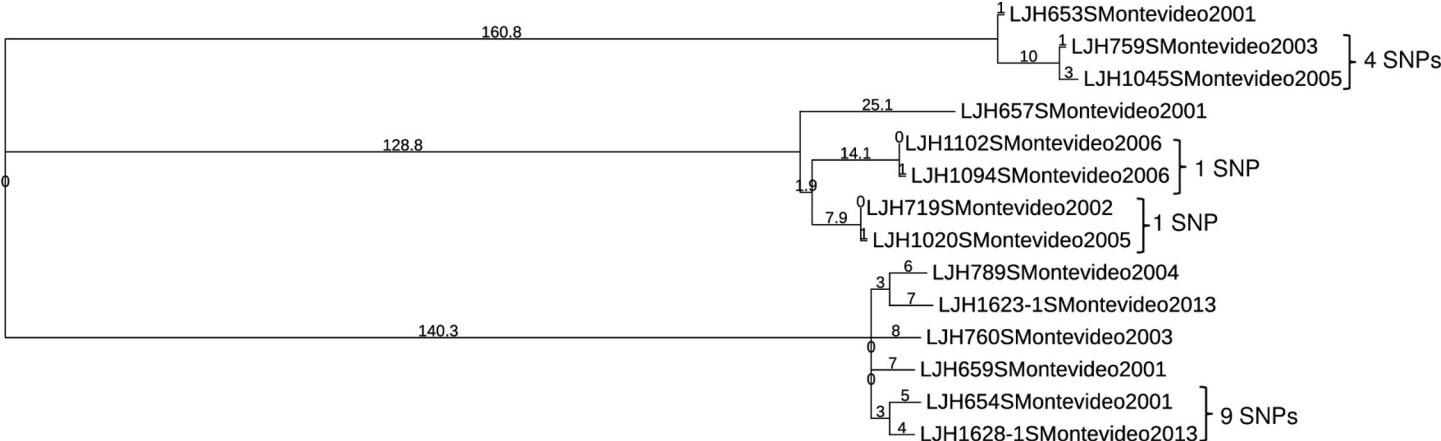

**Fig 3. Phylogenetic tree of *Salmonella* Montevideo generated with the CFSAN SNP Pipeline.** The branch lengths represent the SNP distances among the isolates.

were isolates separated by one or more years (including isolates from 2001 and 2013) that differed by ≤9 SNPs, indicating that they share a common ancestor. *Salmonella* Newport (S6 Table in S1 File) and *Salmonella* Muenchen (S7 Table in S1 File) each clustered into two groups separated by more than 100 SNPs. A small number of SNP differences (<3) between *Salmonella* Newport genomes (S6 Table in S1 File) were identified in isolates retrieved in 2003 (LJH0751), 2006 (LJH1084), 2007 (LJH1133), 2010 (LJH1248, LJH1273 and LJH1278). Nine isolates of *Salmonella* Muenchen were retrieved in 2006; eight of these isolates from six different almond lots had nearly identical genomes, with ≤3 SNP differences (S7 Table in S1 File), and differed from single isolates in 2007 (LJH1135) and 2013 (LJH1661) by ≤8 SNPs. Closely related genomes for single isolates of *Salmonella* Anatum (S8 Table in S1 File) and *Salmonella* Thompson (S9 Table in S1 File) were identified 2 and 1 years apart, respectively. Several *Salmonella* Oranienburg (S10 Table in S1 File) and *Salmonella* Typhimurium (Fig 4) were identified 5 and 3 years apart, respectively. Closely related genomes (≤13 SNPs) were identified for isolates retrieved from separate almond lots during the same year for *Salmonella* serovar Cerro (S11 Table in S1 File), Give (S12 Table in S1 File), Infantis (S13 Table in S1 File), Heidelberg (S14 Table in S1 File) and Tennessee (S15 Table in S1 File). For *Salmonella* serovar Brandenburg (S16 Table in S1 File), Braenderup, Manhattan (S17 Table in S1 File), Mbandaka, and Senftenberg (S18 Table in S1 File), isolates were separated by more than 13 SNPs.

Multiple SNP-based approaches have been developed to analyze the large number of short reads produced by various sequencing platforms [37, 38]. The CFSAN Pipeline was selected because it is less sensitive to coverage changes [37] and has good discriminative power. Its high resolution, however, strongly depends on an appropriate reference genome. In addition to outbreak source attribution, this tool can reveal similarity among isolates and their persistence in food facilities or in the production environment [14, 39].

Differences of more than 100 SNPs were sometimes observed among isolates within each serovar. Survey isolates that were genetically similar (≤13 SNPs) were recovered in multiple years, up to 13 years for *Salmonella* Enteritidis PT 30 and *Salmonella* Montevideo and 10 years for *Salmonella* Newport. Because the survey samples were collected after hulling and shelling and prior to entering a processing facility or storage, the contamination source would have to be at production or harvest (orchard), during postharvest handling (transportation to huller, during storage, or during hulling and shelling), or transportation from huller to processor. Although the exact geographic locations of the almond survey samples were unknown, clonal

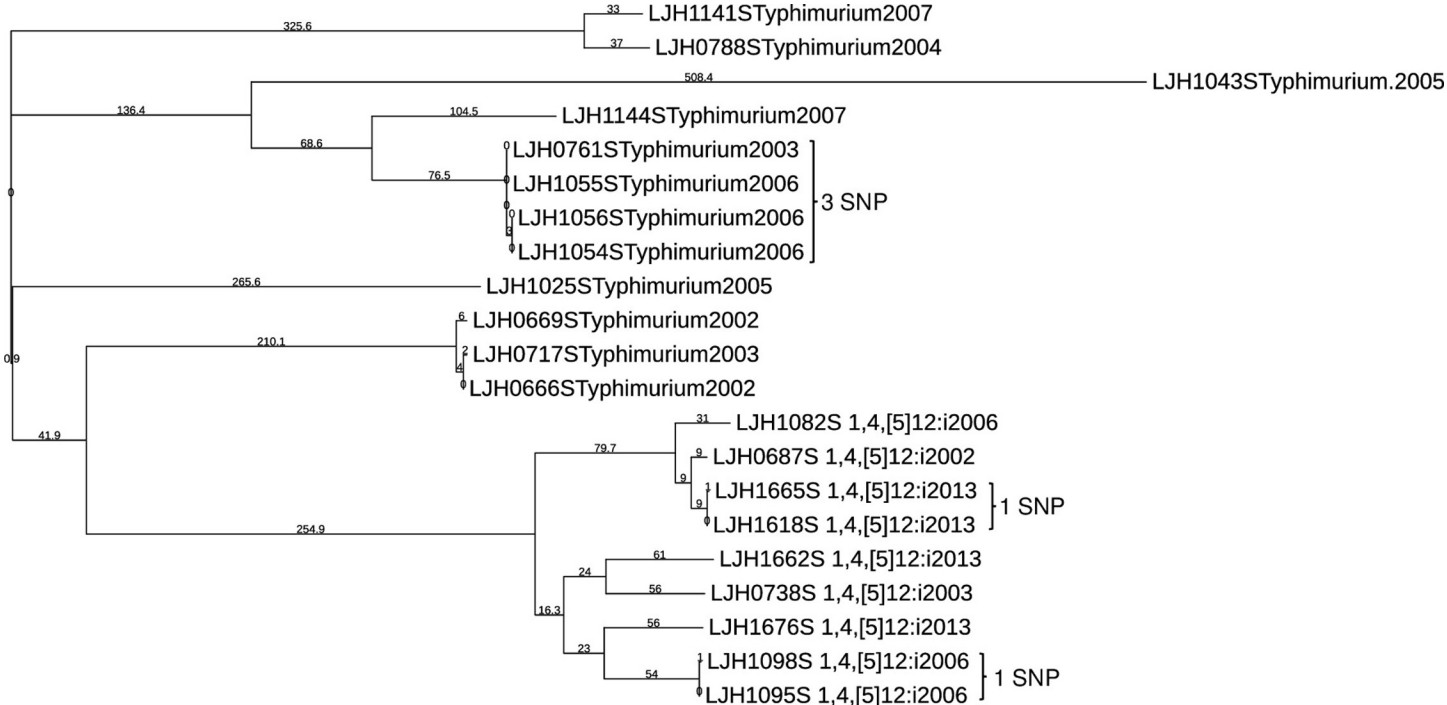

**Fig 4. Phylogenetic tree of *Salmonella* Typhimurium and 1,4,[5],12:i:-.** The branch lengths are representative of the SNP distances among the isolates.

strains of *Salmonella* Enteritidis PT 8 (five; 2005) and *Salmonella* Muenchen (eight; 2006) were isolated from different lots of almonds in the same year.

The diversity of serovars and unique strains within serovars was expected in a survey that likely reflects broad environmental contamination (e.g., almond orchard during production or harvest). Common harvest and postharvest practices may also lead to distribution of *Salmonella* in almonds from geographically diverse orchards that share the same equipment or facilities. At maturity, almonds are shaken to the ground where they dry for several days. They are then harvested by windrowing and sweeping off the orchard floor. The in-hull, inshell almonds are then transported to facilities where the hull and shell are removed. Kernels mix with hulls and shells before sorting, separation, and bulk transportation to processing facilities. Once kernels are delivered to almond processing facilities, commingling of almond lots may occur prior to or during storage. These practices may explain the clusters of *Salmonella* Enteritidis PT 8 and *Salmonella* Muenchen isolated from different lots in 2005 and 2006.

U.S. regulations were implemented in 2007 that require all California-grown almonds sold in North America (U.S., Canada, and Mexico) to be processed with a treatment capable of achieving a minimum 4-log reduction in *Salmonella* [40]. While there have been outbreaks associated with almond-containing products such as blended nut butters, none have been associated with contaminated almonds since 2006, likely due to effective implementation of these regulations [7, 15].

The persistence of *Salmonella* has been described for other pre- and postharvest scenarios [11, 41, 42]. A narrow range of *Salmonella* serovars has been associated with California pistachio outbreaks, outbreak investigations, and industry and retail surveys [41, 43]. Pistachio-associated isolates of *Salmonella* Senftenberg and *Salmonella* Montevideo recovered over multiple years (2009 to 2017) and from multiple facilities differed (within each serovar) by 0 to 31

SNPs, and the authors suggested that the organisms may have established residence in the primary production environment or orchards [41].

## Plasmid prediction and characterization

Plasmids contribute significantly to the emergence and spread of genes encoding AMR, virulence, and other metabolic functions in multiple scales across *Salmonella* serotypes [16, 44]. Plasmid carriage in the *Salmonella* strains under study were assessed by screening and reconstructing the plasmid sequences from assembled genomes using the clustered plasmid reference database-based pipeline [29, 30]. A total of 106 of the 171 *Salmonella* isolates (62%) carried one to seven plasmids (total plasmids 161; Table 1) with sizes of 1,030 bp to 303,322 bp (Fig 5A and S19 Table in S1 File). Using the presence of relaxase and mate-pair formation marker genes and/or *oriT* sequence, most of these plasmids were predicted to be either mobilizable (*n* = 51/161; 32%) or conjugative (*n* = 83/161; 52%) (Fig 5A). In total, 61 plasmid clusters with 27 different plasmid families were identified, with IncFII and IncFIB being the most predominant types in the collection (Fig 5B).

Thirty eight *Salmonella* strains distributed across six serovars (Duisburg, Enteritidis, Lomalinda, Muenchen, Typhimurium, and 1,4,[5],12:i:-) carried plasmids that contained two to seven virulence genes, but no AMR genes (S20 Table in S1 File). The IncFIB-IncFII plasmid family combination was common among these strains (Fig 6), predominantly in *Salmonella* Enteritidis (*n* = 15), Typhimurium (*n* = 8), and 1,4,[5],12:i:- (*n* = 9). Comparative analysis of the plasmid sequences with known plasmids using mash [45] and BLASTn revealed that these plasmids are similar (nucleotide sequence homology = 100%) to plasmid pCFSAN076214_2 (accession number CP033342.1) described previously in *Salmonella* Enteritidis strain ATCC BAA-1045 isolated from raw almonds (also LJH608 in this study) [46] and to plasmid p11-0972.1 (accession number CP039855.1) reported in *S. enterica* serovar 1,4,[5],12:i:- recovered from a human stool sample [47]. IncF plasmids are among the most common plasmids found in *Salmonella* and are reported to carry multiple antibiotic resistance and/or virulence genes, suggesting their role in the dissemination of these genes across *Salmonella* serotypes and Enterobacteriaceae by extension [48, 49].

Thirteen *Salmonella* strains belonging to seven serotypes (Agona, Anatum, Heidelberg, Istanbul, Newport, Typhimurium, and Zerifin) carried plasmids with at least one AMR gene but no virulence genes. The most common among these was within the IncC plasmid family and was associated with six to 10 AMR genes (S20 Table in S1 File). The IncC plasmid identified in the present study was predicted to be conjugative, predominant in *Salmonella* Anatum and *Salmonella* Newport (Fig 6) and was almost indistinguishable (99.99% by BLASTn) from *Salmonella* Anatum plasmid pSAN1-1736 (accession number: CP014658.1) [50]. IncC plasmids are known to be widely distributed across *Salmonella* serotypes from diverse sources and often carry multiple AMR genes [49].

A unique *Salmonella* Kentucky strain LJH1044 carried a large conjugative plasmid (146 Kb) with IncFIB-IncFIC-rep_cluster_2244 plasmid replicons and contained multiple virulence and AMR genes (Fig 6 and S20 Table in S1 File). This plasmid carried three AMR genes encoding resistance to aminoglycoside and tetracycline, a complete *iroBCDEN* operon that encodes salmochelin siderophore [51], and an *iucABCD-iutA* operon that has been described to be associated with aerobactin synthesis essential for virulence and stress response in *Salmonella* [52] (S20 Table in S1 File). Comparative analysis showed that this plasmid had 99.99% nucleotide sequence similarity to plasmid pCVM29188_146 (accession number CP001122.1) reported previously in several *Salmonella* Kentucky strains from poultry in the United States [53, 54].

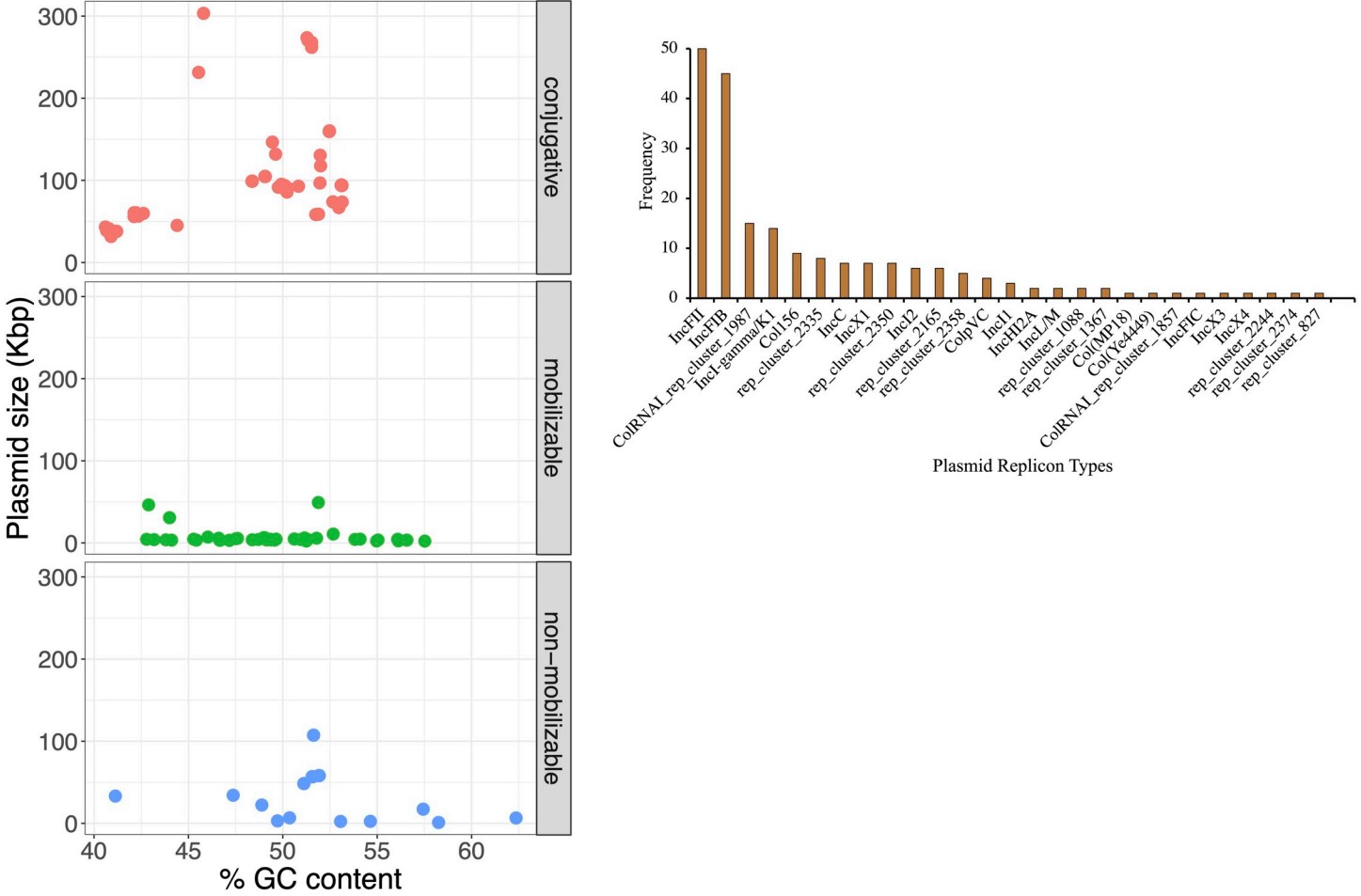

**Fig 5. General features of plasmids identified in *Salmonella* survey isolates included in this study.** (A) Putative plasmids predicted were categorized as mobilizable, conjugative, or non-mobilizable based on the presence or absence of relaxase, mate-pair formation marker genes and/or *oriT* sequence. Each colored circle represents a plasmid-carrying *Salmonella* isolate. (B) Frequency of plasmid replication types.

## Antimicrobial resistance profile

Among the isolates retrieved from the almond survey, a total of 24 AMR genes were identified with the ResFinder and CARD database, which classified into nine different antimicrobial protein groups: aminoglycosides, β-lactams, colistin, fosfomycin, glycopeptide, phenicol, sulfonamide, tetracycline, and trimethoprim. One gene, *aac(6')-Iaa*, which confers resistance to aminoglycosides, was detected in the chromosome of all *Salmonella* subspecies *diarizonae* (3) and *enterica* (165) but not in *arizonae* (3) (Fig 1). The frequency of *Salmonella enterica* subspecies *enterica* isolates that carried an aminoglycoside acetyltransferase *aac(6')-Iaa* has been reported to be high (>95%) in multiple WGS analysis [55–57]. However, in a surveillance study of non-typhoidal *Salmonella enterica*, 11 isolates out of 3,491 (0.3%) showed phenotypic resistance to an aminoglycoside antimicrobial [55].

The other 23 AMR genes were detected in 35 isolates, with *fosA7* being the most common (Fig 1). The gene *fosA7*, which confers resistance to fosfomycin, a broad-spectrum cell wall synthesis inhibitor, was first identified in the chromosome of *Salmonella* Heidelberg isolated from broiler chickens [58]. In the present study, all *Salmonella* Heidelberg isolates (*n* = 6), and some isolates of *Salmonella* serovars Agona (*n* = 2), Meleagris (*n* = 1), Montevideo (*n* = 8),

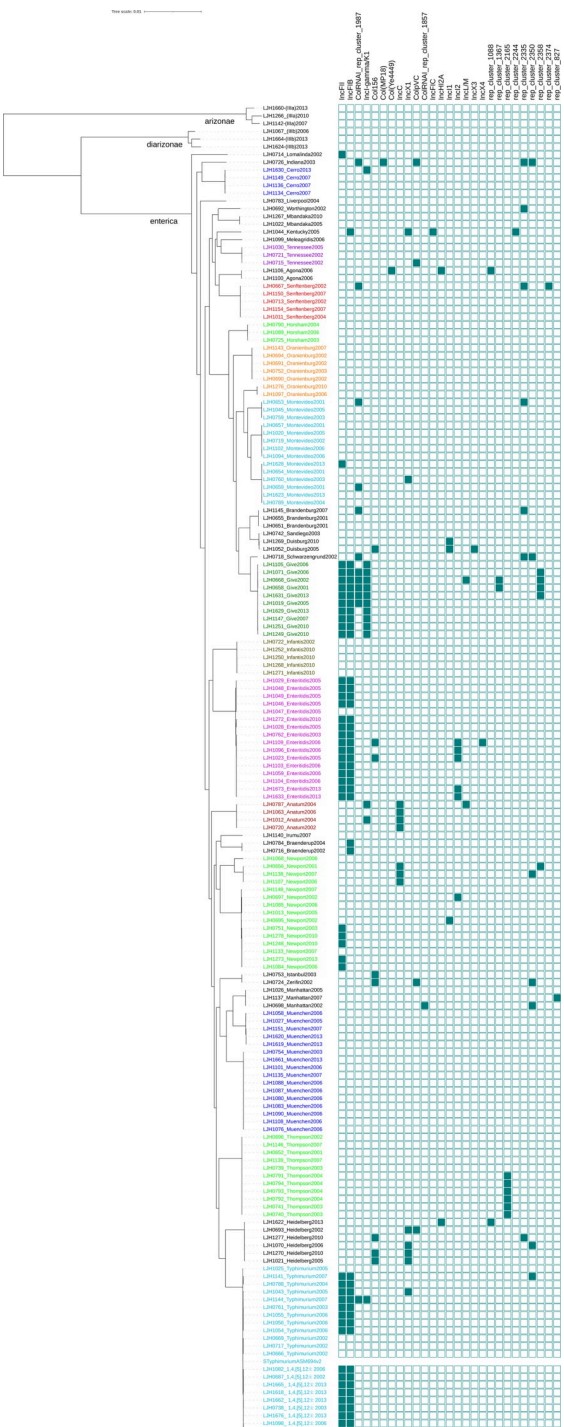

**Fig 6. Heatmap of the distribution of plasmid replication types detected in *Salmonella* isolates.**

Oranienburg (*n* = 2), and Tennessee (*n* = 3), carried the chromosomal *fosA7* gene. Ten amino-glycoside resistance genes were detected in 16 isolates: *aac(6')-Iaa*, *aadA2*, *aadA3*, *aadA7*, *aadA12*, *aadA13*, *ant(2")-Ia*, *aph(3")-IIa*, *aph(3")-Ib*, *aph(6)-lc*, and *aph(6)-Id*. The *bla*$_{CMY-2}$ and *bla*$_{CARB-2}$ genes, which confer resistance to β-lactams, and the *floR* gene, which confers

resistance to phenicol, were found in nine isolates. Three tetracycline efflux resistance genes were identified in 16 isolates: *tetA*, *tetB*, and *tetG*. Sulfisoxazole resistance, encoded by *sul1* or *sul2*, was detected in 11 isolates. The dihydrofolate reductase resistance gene, *dfrA12*, which confers resistance to trimethoprim, was detected in one *Salmonella* Newport isolate. The plasmid-mediated colistin resistance and phosphoethanolamine transferase *mcr*-9.1 gene was detected in one *Salmonella* Agona isolate. Resistance to bleomycin, encoded by the bleomycin-binding protein gene (*ble*), was predicted for one *Salmonella* Heidelberg isolate. All the isolates, except *Salmonella* serovars Enteritidis, Irumu, Lomalinda, Typhimurium, and 1,4,[5],12: i-, contained a missense mutation in *parC* associated with resistance to quinolone. Multiple mutations in the quinolone resistance determining region are usually required to confer resistance to ciprofloxacin, but one mutation confers resistance to nalidixic acid [59]. One *Salmonella* Senftenberg isolate had a point mutation in the 16S rRNA that is thought to confer resistance to spectinomycin.

Most of the predicted AMR genes were identified in a small number (16 out of 171; 9%) of survey isolates and were plasmid encoded in 11 of 16 cases (Fig 1). Multidrug-resistant isolates (putative resistance to at least one antibiotic in three or more drug classes; https://www.cdc. gov/narms/resources/glossary.html) were identified among *Salmonella* serovars Agona

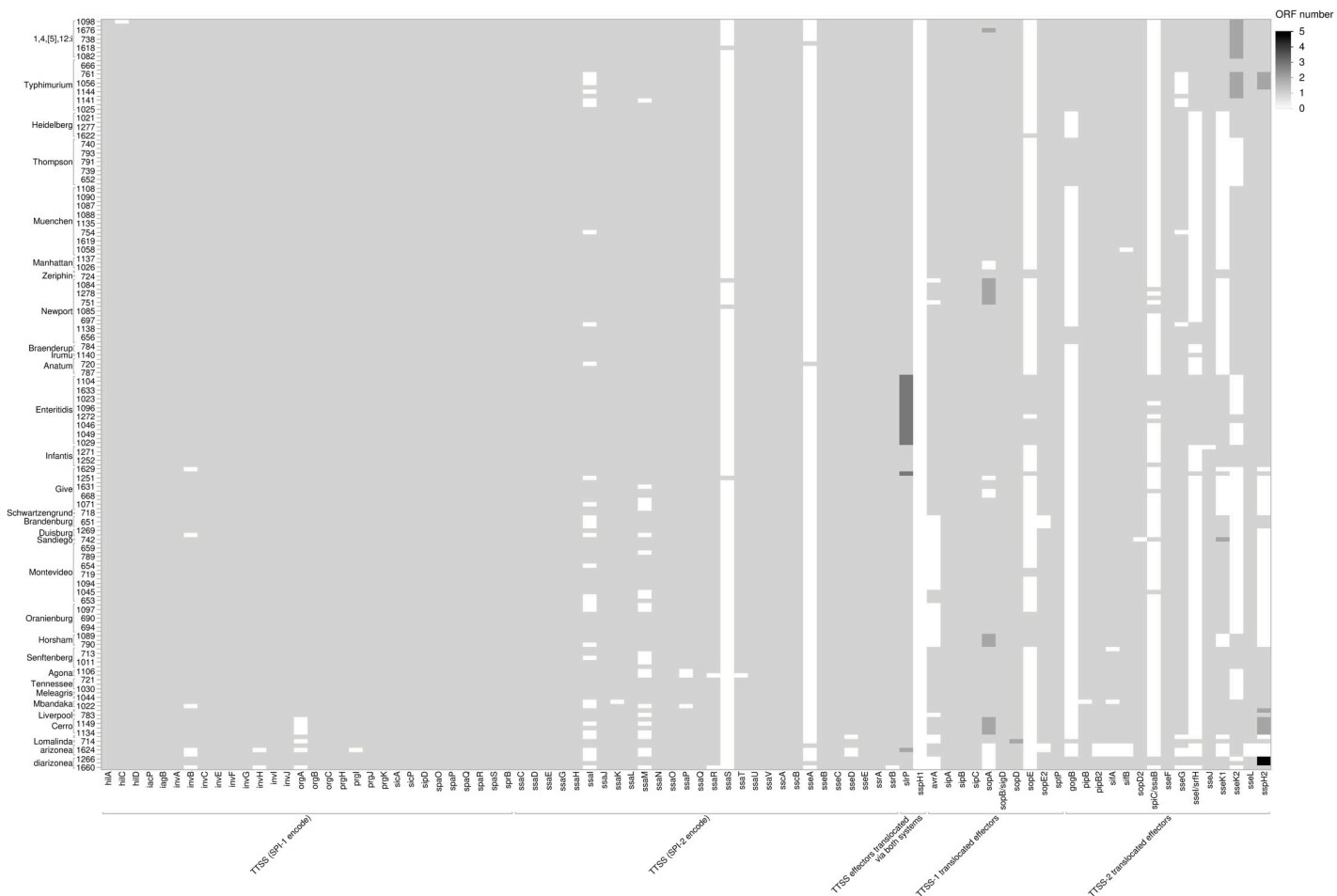

**Fig 7. Heatmap of the distribution of virulence genes encoding type III secretion system (T3SS) across 171 genomes.** The shades of gray represent the number of open reading frames (ORF) that are detected in each putative virulence gene.

(LJH1100 and LJH1106), Anatum (LJH0720, LJH0787, LJH1012, and LJH1063), Heidelberg (LJH1622), Newport (LJH0656, LJH1138, and LJH1107), and Typhimurium (LJH0788 and LJH1141) (Fig 1). Antibiotic resistance profiles by the calibrated dichotomous sensitivity method were determined for *Salmonella* isolated from 2001 through 2005 but not for isolates from 2006, 2007, 2010, and 2013. Ten of the *Salmonella* survey isolates from 2001 to 2005 were resistant to three or more antibiotics [5]. Resistance genotype and phenotype correlated highly for five of these isolates: *Salmonella* Anatum (*n* = 3), *Salmonella* Istanbul (*n* = 1), and *Salmonella* Typhimurium var. Copenhagen (*n* = 1).

## Analysis of virulence factors

*Salmonella* serovars infect a wide range of hosts with different degrees of disease severity, with Enteritidis, Newport, Typhimurium, Javiana, and 1,4,[5],12:i- being significantly more likely to cause illness in humans in the United States [60]. Differences in virulence factors contribute to the severity and outcome of salmonellosis and can be specific to serovars [60]. The Virulence Factor Database (VFDB) was used to detect a total of 303 virulence genes among the 171 *Salmonella* assembled genomes (Figs 7–9). Genes were classified under major virulence factors, including the secretion system, fimbrial and non fimbrial adherence, macrophage

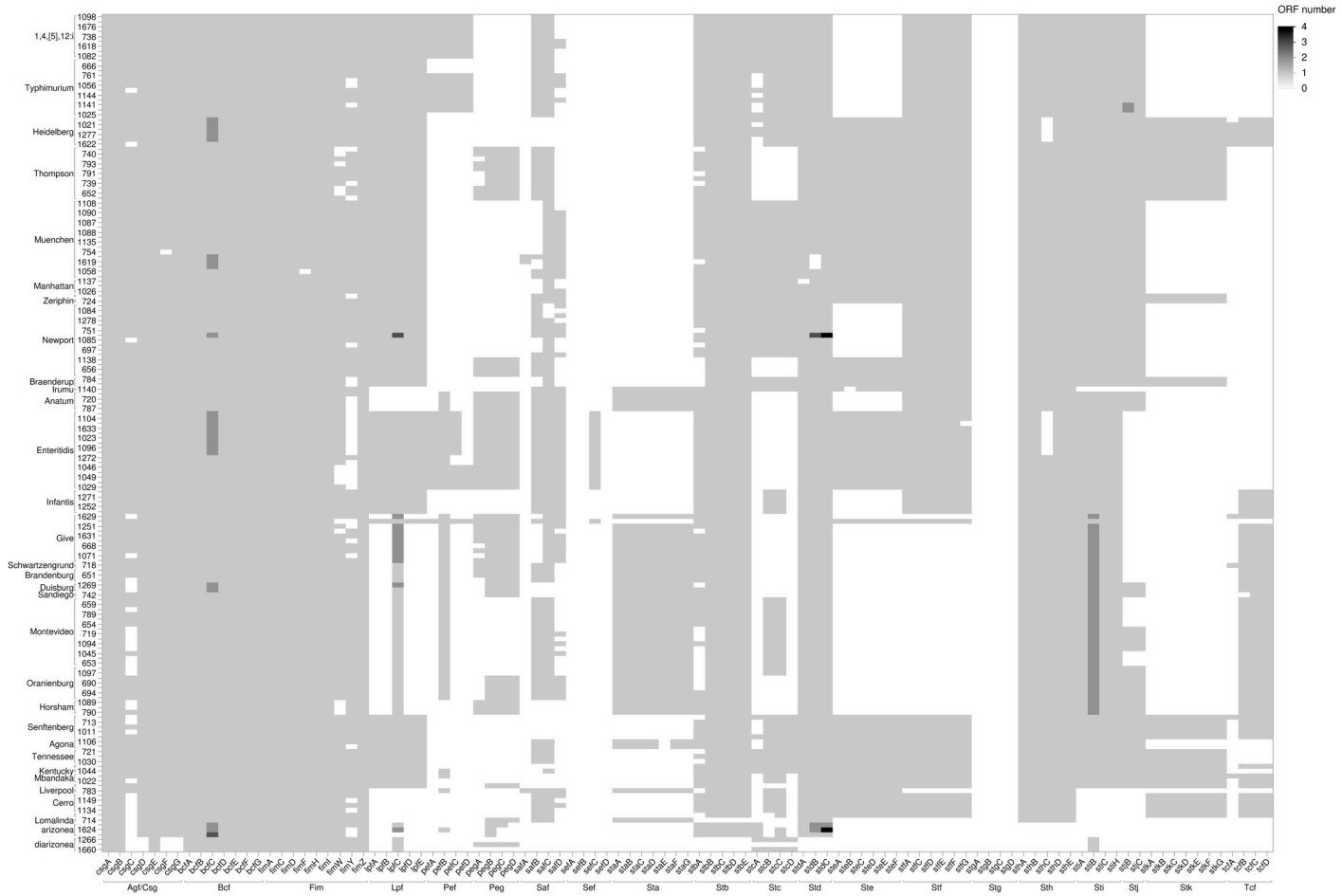

**Fig 8. Heatmap of the distribution of genes encoding the fimbrial operon across 171 genomes.** The shades of gray represent the number of open reading frames (ORF) that are detected in each putative virulence gene.

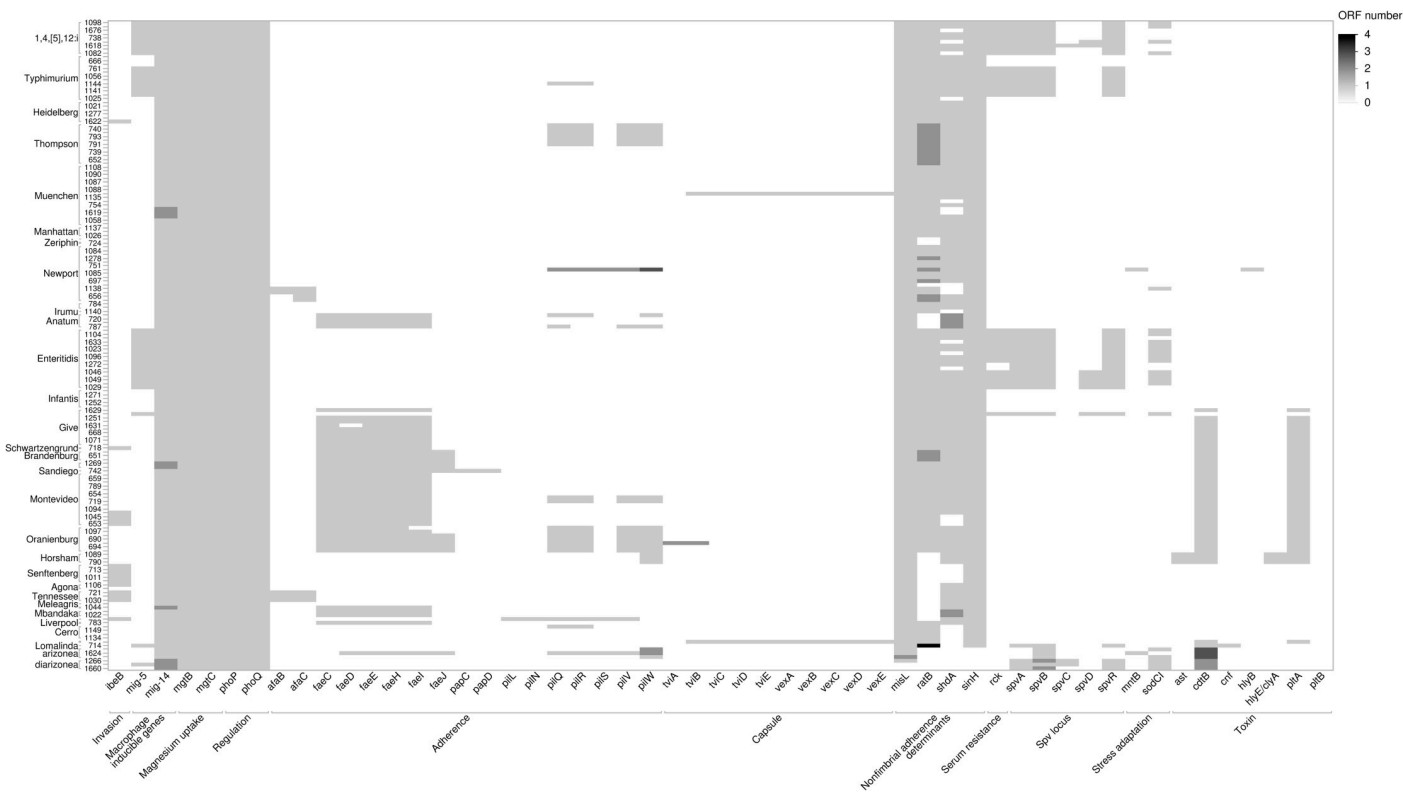

**Fig 9. Heatmap of the distribution of other virulence genes across 171 genomes.** The shades of gray represent the number of open reading frames (ORF) that are detected by in each putative virulence gene.

inducible genes, magnesium uptake, serum resistance, stress proteins, toxins, immune invasion, and two component regulatory systems. The *Salmonella* pathogenicity island 1 (SPI-1) and 2 (SPI-2), responsible for the type III secretion system, are ubiquitous in *S. enterica* subsp. *enterica* [60] and were common to all 171 survey isolates (Fig 7).

Among the fimbrial adherence factors, the genes encoding the curli fimbriae, *csgA*, *csgB*, and *csgE*, the *bcfABCDEFG* operon, and the genes that encode type 1 fimbriae, *fimA*, *fimB*, *fimC*, *fimD*, *fimH*, *fimI*, *fimW*, and *fimZ*, were also present in all the isolates (Fig 8). The two-component regulatory system *phoP-PhoQ* genes and the magnesium uptake genes, *mgtB-mgtC*, (part of SPI-3) were present in all the isolates (Fig 9). All the isolates had the microphage inducible gene, *mig-14*, but *mig-5* was mainly detected in isolates of *Salmonella* serovars Enteritidis, Typhimurium, and 1,4,[5]12:I:- (Fig 9).

The typhoid toxin genes, *cdtB* and *pltA*, originally identified in serotype Typhi, were found in all *Salmonella* serovar Brandenburg, Duisburg, Horsham, Montevideo, Sandiego, and Schwartzengrund isolates and in nine of 10 *Salmonella* Oranienburg isolates (Fig 9). However, *pltB*, required for forming holotoxin, was not present in any of these isolates. In *Salmonella* Horsham isolates, two genes were identified as homologs of the enterotoxin hemolysin genes of *Escherichia coli* (*hylE/clyA*) (Fig 9).

A virulence plasmid that harbored a combination of *pef*, *rck*, and *spv* virulence genes was identified in 23% of the isolates (S20 Table in S1 File). The assembly of the major Pef fimbriae depends on the *pefBACDorf5orf6* operon, which encodes PefA fimbriae subunit, PefC usher protein, and the pefD periplasmic chaperone [61]. Eight of 10 *Salmonella* Typhimurium and

all nine *Salmonella* 1,4,[5]12:I:- isolates carried a plasmid encoding *pefA*, *pefB*, *pefC*, and *pefD* genes (Fig 8 and S20 Table in S1 File).

The *spv* genes play a role in suppression of the innate immune response and are often associated with invasive disease and increased virulence [62, 63]. The *spv* genes were detected in all *Salmonella* serovar Lomalinda, Enteritidis, and 1,4,[5]12:I:- isolates, in eight of 10 *Salmonella* Typhimurium isolates, and one of 10 *Salmonella* Give isolates (Fig 9 and S20 Table in S1 File). The *rck* gene, which provides protection against the complement-mediated immune response of the host [64], was found in one of 10 *Salmonella* Give isolates, in all *Salmonella* Enteritidis PT 30 and PT 8 isolates, in eight of 10 *Salmonella* Typhimurium isolates, and all *Salmonella* 1,4,[5]12:I;- isolates (Fig 9).

This study provides one of the first in-depth longitudinal characterizations of *Salmonella* strains isolated from a single product (almonds) or production environment (almond orchard), in a single geographical region (Central California). This isolate collection is important for understanding *Salmonella* populations in a significant food production region of the United States. Several clonal strains of *Salmonella* were isolated over multiple years, adding to a growing body of evidence that enteric pathogens may persist over long periods of time (years) in agricultural environments and in postharvest or food processing facilities (https://www.cdc.gov/ncezid/dfwed/outbreak-response/rep-strains.html) [41, 42]).

## Supporting information

**S1 File.**
(XLSX)

## Acknowledgments

We thank Sylvia Yada for editing the manuscript and Vanessa Lieberman for technical assistance.

## Author Contributions

**Conceptualization:** Anne-laure Moyne, Linda J. Harris.

**Data curation:** Anne-laure Moyne, Irena Kukavica-Ibrulj.

**Formal analysis:** Anne-laure Moyne, Opeyemi U. Lawal, Jeff Gauthier, Irena Kukavica-Ibrulj, Marianne Potvin.

**Funding acquisition:** Lawrence Goodridge, Roger C. Levesque, Linda J. Harris.

**Investigation:** Anne-laure Moyne.

**Methodology:** Anne-laure Moyne, Opeyemi U. Lawal, Jeff Gauthier, Irena Kukavica-Ibrulj, Marianne Potvin, Roger C. Levesque.

**Project administration:** Linda J. Harris.

**Resources:** Lawrence Goodridge.

**Validation:** Anne-laure Moyne.

**Visualization:** Anne-laure Moyne, Opeyemi U. Lawal.

**Writing – original draft:** Anne-laure Moyne, Opeyemi U. Lawal, Linda J. Harris.

**Writing – review & editing:** Anne-laure Moyne, Opeyemi U. Lawal, Lawrence Goodridge, Roger C. Levesque, Linda J. Harris.

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
