## [Decision Letter · Decision Letter 0]

27 Jun 2023

PONE-D-23-14605Genetic diversity of * Salmonella enterica * isolated from raw California almonds and from an almond orchard over 13 YearsPLOS ONE

Dear Dr. Harris,

Thank you for submitting your manuscript to PLOS ONE. After careful consideration, we feel that it has merit but does not fully meet PLOS ONE’s publication criteria as it currently stands. Therefore, we invite you to submit a revised version of the manuscript that addresses the points raised during the review process.

We look forward to receiving your revised manuscript.

Kind regards,

Iddya Karunasagar

Academic Editor

PLOS ONE

Journal Requirements:

   "This research was supported by the Almond Board of California Grant 18-HarrisL536 AQFSS-01. The DNA Technologies and Expression Analysis Core at UC Davis Genome Center is supported by NIH shared Instrumentation Grant 1S10OD010786-01. L. Goodridge and RC Levesque were funded by Genome Canada and Genome Québec. We thank Sylvia Yada for editing the manuscript and Vanessa Lieberman for technical assistance."

   "LJH 18-HarrisL-AQFSS-01 Almond Board of California (https://www.almonds.com) 

LG/RCL Genome Canada (https://genomecanada.ca) and Genome Quebec (https://www.genomequebec.com/en/home/)

Additional Editor Comments:

Please see the reviewer comments. Please revise addressing all comments point by point

Reviewers' comments:

Reviewer's Responses to Questions

**Comments to the Author**

1. Is the manuscript technically sound, and do the data support the conclusions?

Reviewer #1: Yes

2. Has the statistical analysis been performed appropriately and rigorously? 

Reviewer #1: N/A

3. Have the authors made all data underlying the findings in their manuscript fully available?

Reviewer #1: Yes

4. Is the manuscript presented in an intelligible fashion and written in standard English?

Reviewer #1: Yes

5. Review Comments to the Author

Reviewer #1: PONE-D-23-14605

Genetic diversity of Salmonella enterica isolated from raw California almonds and from an almond orchard over 13 years

This study compared the whole genome sequences of 171 Salmonella isolates that included 30 isolates of S. Enteritidis PT30 involved in almond-associated outbreaks. SNP analysis, serovar identification, antimicrobial resistance genes, plasmid types and virulence genes were determined. The main goal of this study was to see if S. Enteritidis PT30 outbreak isolates could be distinguished from similar environmental isolates.

The study emphasizes the discriminatory power of WGS over traditional genotyping methods. The study identified clonal isolates of Salmonella Enteritidis and Salmonella Montevideo. The persistence of clonal strains of certain serovars suggest contamination of post-harvest processing areas with these strains.

1. The manuscript is straightforward presentation of the WGS comparison. The study assumes significance in few contexts; that Salmonella Enteritidis PT 30 isolates belong to a long time line of over 13 years, comprise of outbreak strains and their environmental counterparts, and as the result suggest, these strains were phylogenetically closely related.

The study aimed to solve three important issues with the traditional genotyping methods as stated in the introduction section and listed below.

2. Lines 80-82:. “However, neither PFGE nor MVLA could discriminate among the Salmonella Enteritidis PT 30 isolates associated with the 2001 and 2006 almond-associated outbreaks”

Now, the WGS analysis clustered (lines 275-276) clustered them together in one group with ≤18 SNPs. Is this not similar to PFGE results? Or does a difference of 18 SNPs make them different from each other?

3. Lines 82-84: Salmonella Enteritidis PT 30 almond associated outbreak strains could not be distinguished from epidemiologically unrelated Salmonella Enteritidis PT 30 clinical strains included in the study

Again, the fact that the WGS analysis put them together in one group suggests the same thing.

4. Lines 87-89: Environmental Salmonella Enteritidis PT 30 isolates, collected between 2001 and 2006 from one of the 2001 outbreak-associated orchards, clustered in two groups based on the separation by PFGE of their XbaI-digested DNA

As could be seen from the results in lines 274-276, one survey isolate formed a separate group.

5. Lines 294-295: With less than ≤18 SNPs, are they still termed “closely related strains”. In a �4 Mb genome sequence, what would be the NGS contribution to SNPs? Although this depends on the depth of sequencing, still there would be contribution from the sequencing itself.

6. Although AMR genes were identified from the WGS, these are not correlated with the antibiotic resistance phenotype of isolates.

7. Only two pathogenicity islands (SPI-1 & SPI-2) were identified in WGS. What about the rest?

6. PLOS authors have the option to publish the peer review history of their article (what does this mean?). If published, this will include your full peer review and any attached files.

Reviewer #1: No

---

## [Author Response · Author response to Decision Letter 0]

4 Aug 2023

PONE-D-23-14605

Genetic diversity of Salmonella enterica isolated over 13 years from raw California almonds and from an almond orchard

Editor and Reviewer comments are in italics. Author responses are not in italics. 

Editor c1. A rebuttal letter that responds to each point raised by the academic editor and reviewer(s). You should upload this letter as a separate file labeled 'Response to Reviewers'.

Response: Done

Editor c2. A marked-up copy of your manuscript that highlights changes made to the original version. You should upload this as a separate file labeled 'Revised Manuscript with Track Changes'.

Response: Done.

Editor c3. An unmarked version of your revised paper without tracked changes. You should upload this as a separate file labeled 'Manuscript'.

Response: Done.

Journal Requirements:

JR 1. Please ensure that your manuscript meets PLOS ONE's style requirements, including those for file naming. The PLOS ONE style templates can be found at 

JR 2. Thank you for stating the following in the Acknowledgments Section of your manuscript: "This research was supported by the Almond Board of California Grant 18-HarrisL536 AQFSS-01. The DNA Technologies and Expression Analysis Core at UC Davis Genome Center is supported by NIH shared Instrumentation Grant 1S10OD010786-01. L. Goodridge and RC Levesque were funded by Genome Canada and Genome Québec. We thank Sylvia Yada for editing the manuscript and Vanessa Lieberman for technical assistance."

We note that you have provided funding information that is not currently declared in your Funding Statement. However, funding information should not appear in the Acknowledgments section or other areas of your manuscript. We will only publish funding information present in the Funding Statement section of the online submission form. Please remove any funding-related text from the manuscript and let us know how you would like to update your Funding Statement. Currently, your Funding Statement reads as follows: "LJH 18-HarrisL-AQFSS-01 Almond Board of California (https://www.almonds.com) LG/RCL Genome Canada (https://genomecanada.ca) and Genome Quebec (https://www.genomequebec.com/en/home/). The funders had no role in study design, data collection and analysis, decision to publish, or preparation of the manuscript."

Response: We have deleted the funding information from the Acknowledgement section in the manuscript. 

Amended Funding Statement: “LJH 18-HarrisL-AQFSS-01 Almond Board of California (https://www.almonds.com), LG/RCL Genome Canada (https://genomecanada.ca) and Genome Quebec (https://www.genomequebec.com/en/home/), NIH shared Instrumentation Grant 1S10OD010786 (https://orip.nih.gov/construction-and-instruments/s10-instrumentation-programs). The funders had no role in study design, data collection and analysis, decision to publish, or preparation of the manuscript."

JR 3. We note that you have stated that you will provide repository information for your data at acceptance. Should your manuscript be accepted for publication, we will hold it until you provide the relevant accession numbers or DOIs necessary to access your data. If you wish to make changes to your Data Availability statement, please describe these changes in your cover letter and we will update your Data Availability statement to reflect the information you provide.

Response: We included all relevant accession numbers in the manuscript and have now released those data.

JR 4. Please review your reference list to ensure that it is complete and correct. If you have cited papers that have been retracted, please include the rationale for doing so in the manuscript text, or remove these references and replace them with relevant current references. Any changes to the reference list should be mentioned in the rebuttal letter that accompanies your revised manuscript. If you need to cite a retracted article, indicate the article’s retracted status in the References list and also include a citation and full reference for the retraction notice.

Response: Reference list has been reviewed and is complete and correct.

Reviewer #1: 

This study compared the whole genome sequences of 171 Salmonella isolates that included 30 isolates of S. Enteritidis PT30 involved in almond-associated outbreaks. SNP analysis, serovar identification, antimicrobial resistance genes, plasmid types and virulence genes were determined. The main goal of this study was to see if S. Enteritidis PT30 outbreak isolates could be distinguished from similar environmental isolates.

The study emphasizes the discriminatory power of WGS over traditional genotyping methods. The study identified clonal isolates of Salmonella Enteritidis and Salmonella Montevideo. The persistence of clonal strains of certain serovars suggest contamination of post-harvest processing areas with these strains.

The manuscript is straightforward presentation of the WGS comparison. The study assumes significance in few contexts; that Salmonella Enteritidis PT 30 isolates belong to a long timeline of over 13 years, comprise of outbreak strains and their environmental counterparts, and as the result suggest, these strains were phylogenetically closely related. 

The study aimed to solve three important issues with the traditional genotyping methods as stated in the introduction section and listed below.

Response: Thank you for your comments and manuscript overview. As stated in the last paragraph in the introduction, the objective of the study was a comparative genomic analysis of almond-related Salmonella isolates. The study did not aim to solve issues with traditional genotyping methods. However, we have addressed each comment related to statements in the introduction below.

R1c1a Lines 80-82: “However, neither PFGE nor MVLA could discriminate among the Salmonella Enteritidis PT 30 isolates associated with the 2001 and 2006 almond-associated outbreaks”. Now, the WGS analysis clustered (lines 275-276) clustered them together in one group with ≤18 SNPs. Is this not similar to PFGE results? Or does a difference of 18 SNPs make them different from each other?

R1c1b Lines 82-84: Salmonella Enteritidis PT 30 almond associated outbreak strains could not be distinguished from epidemiologically unrelated Salmonella Enteritidis PT 30 clinical strains included in the study. Again, the fact that the WGS analysis put them together in one group suggests the same thing.

R1c1c Lines 87-89: Environmental Salmonella Enteritidis PT 30 isolates, collected between 2001 and 2006 from one of the 2001 outbreak-associated orchards, clustered in two groups based on the separation by PFGE of their XbaI-digested DNA. As could be seen from the results in lines 274-276, one survey isolate formed a separate group.

Response: The statement regarding the PFGE or MVLA was specific to clinical isolates of Salmonella Enteritidis PT 30 which could not be distinguished from each other using PFGE or MVLA (refers to Parker et al. [8]). 

With one exception (LJH0762), we were also unable to distinguish the Salmonella Enteritidis PT 30 isolates assessed in the present paper by WGS (beyond 18 SNP differences). Our analysis consisted of 55 isolates: eight almond survey isolates, 30 orchard isolates and 16 outbreak isolates. We did not include some of the non-almond associated clinical isolates that were evaluated in the Parker et al [8] study and thus are unable to know if WGS could distinguish among all the isolates used in that study.

To clarify, we made minor modification to the concluding paragraph of the introduction to specify that 171 isolates (45 serovars) were evaluated (not just Salmonella Enteritidis PT 30). We provided additional information under “Genetic Distance Within Each Serovar” (copied below) to highlight the number of Salmonella Enteritidis PT 30 isolates that were assessed. We added a new supplemental table (S5 Table) that complements Figure 2 and provides more granularity to SNP differences. We also further clarified that almond isolates from 2001 to 2013 differed from the 2001 outbreak-associated almond isolate LJH0608 (from almonds harvested in 2000) by 2 to 13 SNPs and that this represents isolates collected over a span of 14 years.

We also modified the following sentence to remove the statement “with higher resolution”:

“The persistence of Salmonella Enteritidis PT 30 in an almond orchard over 6 years was reported previously [2]. The SNP analysis confirmed the PFGE results obtained for these isolates.”

The reviewer’s statement (R1c1c) regarding environmental Salmonella isolates was related to orchard isolates (refers to Uesugi et al., 2006 [2]). The comment refers to only the environmental samples retrieved from the outbreak-associated almond orchard. The orchard isolates from 2001 to 2006 differed by 0 to 12 SNPs in our analyses, which is now more clearly stated in the modified results section (copied below). We believe that the changes outlined above adequately address the reviewer’s comments. 

“The genomes of Salmonella Enteritidis PT 30 recovered from survey almonds (eight isolates (Table 1): LJH0762 [2003], LJH1023 [2005], LJH1104 [2006], LJH1059 [2006], LJH1096 [2006], LJH1109 [2006], LJH1633 [2013], LJH1673 [2013]), the 2001 outbreak-associated orchard (30 isolates; S2 Table), and a 2001 outbreak-associated almond isolate (LJH0608) were compared to Salmonella Enteritidis PT 30 genomes of clinical isolates from almond outbreaks in 2001 (12 isolates) and 2006 (four isolates) (S3 Table). 

Salmonella Enteritidis PT30 isolates formed two clusters (Fig 2). One consisted of a single survey isolate (LJH0762), recovered in 2003, that differed from LJH0608 by 48 SNPs (Fig 2, S5 Table). All other survey and clinical isolates (n = 38) clustered in a single group with LJH0608 that differed from each other by ≤18 SNPs (Fig 2) indicating that the isolates are from a common origin. Almond isolates from 2001 to 2013 had two to 13 SNP differences compared with the 2001 outbreak-associated almond isolate Salmonella Enteritidis PT 30 LJH0608 (Fig 2, S5 Table). Although this isolate was recovered from recalled almonds in 2001, the almonds were harvested in the fall of 2000 [1], a span of 14 years (2000–2013). The orchard isolates from 2001 to 2006 differed by 0 to 12 SNPs within their genomes and by 3 to 13 SNPs with the clinical genomes. The SNP differences ranged from zero to eight within the 12 clinical isolates from the 2001 outbreak and from one to 13 within the four clinical isolates from the 2006 outbreak. Among the clinical isolates from 2001 and 2006, the SNP differences ranged from four to 13.” 

R1c2 Lines 294-295: With less than ≤18 SNPs, are they still termed “closely related strains”. In a �4 Mb genome sequence, what would be the NGS contribution to SNPs? Although this depends on the depth of sequencing, still there would be contribution from the sequencing itself.

Response: There isn’t a standard cutoff for SNP differences with respect to strain separation (Brown et al. 2019) (9). Parameters defined in the CFSAN pipeline for filtering the SNPs should minimize the contribution of NGS to SNPs. The CFSAN pipeline constructs a high-quality SNP matrix for closely related sequences (<100 SNP differences) with a higher recovery rate of SNPs for datasets with 100x coverage compared to 20x, as described by Davis et al., 2015 (12). Our genomes were sequenced with an average coverage of 50x. 

We believe that modifications to this section (as described above) have addressed this question.

R1c3 Although AMR genes were identified from the WGS, these are not correlated with the antibiotic resistance phenotype of isolates.

Response: Throughout the section on AMR we specifically note the WGS refers to genotype and not phenotype. However, in the last paragraph of the “Antimicrobial resistance profile” section we discuss the information that is available on genotypic vs phenotypic expression of AMR genes in some of the isolates associated with the study (copied below). We believe this addresses the reviewer concerns and have not made changes to the manuscript.

“Antibiotic resistance profiles by the calibrated dichotomous sensitivity method were determined for Salmonella isolated from 2001 through 2005 but not for isolates from 2006, 2007, 2010, and 2013. Ten of the Salmonella survey isolates from 2001 to 2005 were resistant to three or more antibiotics [5]. Resistance genotype and phenotype correlated highly for five of these isolates: Salmonella Anatum (n = 3), Salmonella Istanbul (n = 1), and Salmonella Typhimurium var. Copenhagen (n = 1).”

R1c4 Only two pathogenicity islands (SPI-1 & SPI-2) were identified in WGS. What about the rest?

Response: SPI-1 and SPI-2 were present in all the isolates and we made note of this. Genes from other known pathogenicity islands were present but we did not assess the presence of complete islands in every isolate. The magnesium uptake genes (mgtB-mgtC) were present in all isolates (line 502) and are part of SPI-3. The hlyE hemolysin (lines 511) is part of SPI-18 and was identified in S. Horsham.

---

## [Decision Letter · Decision Letter 1]

22 Aug 2023

Genetic diversity of * Salmonella enterica * isolated over 13 years from raw California almonds and from an almond orchard

PONE-D-23-14605R1

Dear Dr. Harris,

We’re pleased to inform you that your manuscript has been judged scientifically suitable for publication and will be formally accepted for publication once it meets all outstanding technical requirements.

Kind regards,

Iddya Karunasagar

Academic Editor

PLOS ONE

Additional Editor Comments (optional):

All reviewer comments have been addressed.

Reviewers' comments:

Reviewer's Responses to Questions

**Comments to the Author**

1. If the authors have adequately addressed your comments raised in a previous round of review and you feel that this manuscript is now acceptable for publication, you may indicate that here to bypass the “Comments to the Author” section, enter your conflict of interest statement in the “Confidential to Editor” section, and submit your "Accept" recommendation.

Reviewer #1: All comments have been addressed

2. Is the manuscript technically sound, and do the data support the conclusions?

Reviewer #1: Yes

3. Has the statistical analysis been performed appropriately and rigorously? 

Reviewer #1: Yes

4. Have the authors made all data underlying the findings in their manuscript fully available?

Reviewer #1: Yes

5. Is the manuscript presented in an intelligible fashion and written in standard English?

Reviewer #1: Yes

6. Review Comments to the Author

Reviewer #1: The manuscript has been sufficiently revised, and all my querries in the previous review have been addresssed. The authors have made relevant changes in the revised manuscript.

7. PLOS authors have the option to publish the peer review history of their article (what does this mean?). If published, this will include your full peer review and any attached files.

Reviewer #1: No

---

## [Editor Report · Acceptance letter]

29 Aug 2023

PONE-D-23-14605R1 

Genetic diversity of *Salmonella enterica* isolated over 13 years from raw California almonds and from an almond orchard 

Dear Dr. Harris:

I'm pleased to inform you that your manuscript has been deemed suitable for publication in PLOS ONE. Congratulations! Your manuscript is now with our production department. 

Kind regards, 

on behalf of

Dr. Iddya Karunasagar 

Academic Editor

PLOS ONE